

# Entanglement measures in a nonequilibrium steady state: Exact results in one dimension

Shachar Fraenkel⋆ and Moshe Goldstein

Raymond and Beverly Sackler School of Physics and Astronomy,
Tel-Aviv University, Tel Aviv 6997801, Israel

⋆ shacharf@mail.tau.ac.il

## Abstract

Entanglement plays a prominent role in the study of condensed matter many-body systems: Entanglement measures not only quantify the possible use of these systems in quantum information protocols, but also shed light on their physics. However, exact analytical results remain scarce, especially for systems out of equilibrium. In this work we examine a paradigmatic one-dimensional fermionic system that consists of a uniform tight-binding chain with an arbitrary scattering region near its center, which is subject to a DC bias voltage at zero temperature. The system is thus held in a current-carrying nonequilibrium steady state, which can nevertheless be described by a pure quantum state. Using a generalization of the Fisher-Hartwig conjecture, we present an exact calculation of the bipartite entanglement entropy of a subsystem with its complement, and show that the scaling of entanglement with the length of the subsystem is highly unusual, containing both a volume-law linear term and a logarithmic term. The linear term is related to imperfect transmission due to scattering, and provides a generalization of the Levitov-Lesovik full counting statistics formula. The logarithmic term arises from the Fermi discontinuities in the distribution function. Our analysis also produces an exact expression for the particle-number-resolved entanglement. We find that although to leading order entanglement equipartition applies, the first term breaking it grows with the size of the subsystem, a novel behavior not observed in previously studied systems. We apply our general results to a concrete model of a tight-binding chain with a single impurity site, and show that the analytical expressions are in good agreement with numerical calculations. The analytical results are further generalized to accommodate the case of multiple scattering regions.



# 1 Introduction

Soon after the nascence of quantum mechanics, entanglement was recognized as a unique attribute of quantum systems. Yet only over the past few decades has its consequential nature truly been acknowledged, in particular in the realm of many-body physics [1–3]. Entanglement is now routinely harnessed to detect quantum phase transitions, both in and out of equilibrium [4–9]; to characterize long-range correlations in contexts such as dynamics [10–14] or topological order [15–17]; and to analyze the capabilities and limitations of simulation methods [18–20].

The prospect of promoting our understanding of nonequilibrium quantum systems through the investigation of their entanglement properties is especially intriguing. The study of quantum many-body phenomena out of equilibrium has shown promising progress in recent years, propelled by the development of suitable quantum simulation platforms in cold atom systems [21–25]. But while the theoretical understanding of quantum many-body nonequilibrium has substantially progressed [26, 27], rigorous analytical results are still rare. Entanglement measures have established a route for producing such results [28–37], a route which this work seeks to advance.

Entanglement entropy [1, 2] is a measure usually employed to quantify entanglement within a many-body system in a pure state, represented by a density matrix $\rho$. Given a bipartition of the total system into subsystems $A$ and $B$, one obtains the reduced density matrix (RDM) of subsystem $A$ by tracing out the degrees of freedom associated with subsystem $B$,

$\rho_A = \text{Tr}_B[\rho]$. The von-Neumann entanglement entropy (vNEE) is then defined as

$$\mathcal{S} = -\text{Tr}[\rho_A \ln \rho_A]. \tag{1}$$

Additionally, we denote the $n$th moment of the RDM as

$$Z_n = \text{Tr}[\rho_A^n], \tag{2}$$

and refer to it as the Rényi moment of order $n$. Note that this is slightly different than the Rényi entropy, $S_n = \frac{1}{1-n} \ln(\text{Tr}[\rho_A^n])$. The von-Neumann entropy and the Rényi moments are related through

$$\mathcal{S} = -\lim_{n \to 1} \partial_n Z_n. \tag{3}$$

Between these two measures of bipartite entanglement, the vNEE constitutes the more rigorous one in and of itself [1, 2]. Nevertheless, Rényi entropies may be used to provide lower bounds to the vNEE and to reconstruct the full entanglement spectrum [17, 38, 39], and are also accessible to direct experimental measurement [40, 41].

Of prime importance is the way in which the vNEE scales with the size of the subsystem in question. This scaling law is considered to be a significant classification criterion, distinguishing between typical phases of condensed matter systems [3]. A prominent example is the renowned area law, under which the vNEE scales linearly with the area of the subsystem's boundary: $\mathcal{S} \sim cL^{d-1}$, with $L$ being a typical linear dimension of the subsystem, $d$ being the spatial dimension and $c$ being a constant [42]. The area law generally applies to ground states of gapped local Hamiltonians [43, 44], as well as to excited eigenstates of many-body-localized systems [14, 45, 46], and has generated particular interest due to the fact that states obeying the area law admit an efficient tensor-network representation [20, 44, 47–49]. Ground states of gapless systems (and specifically critical systems) with a finite and sharp Fermi surface tend to violate the area law by a logarithmic correction, $\mathcal{S} \sim cL^{d-1} \ln L$ [6, 7, 50, 51]. On the other hand, a volume-law scaling – i.e., an extensive scaling of the vNEE, $\mathcal{S} \sim cL^d$ – is widely observed in highly-excited states of local thermalizing Hamiltonians [52–55] and in ground states of non-local Hamiltonians [52].

An additional tool in the analysis of many-body entanglement, that has lately come into increased awareness, is the symmetry- or charge-resolved entanglement entropy [56–60]. Given an additive quantity $Q = Q_A + Q_B$ that is globally conserved in the total system (e.g., particle number), the RDM $\rho_A$ derived for a pure eigenstate of the Hamiltonian turns out to be block-diagonal with respect to the eigenbasis of $Q_A$, $\rho_A = \oplus_{Q_A} \rho_A^{(Q_A)}$. This suggests that entanglement entropies may be calculated for each block separately, thus resolving the total Rényi moments and vNEE to sums over contributions from symmetry sectors:

$$Z_n = \sum_{Q_A} Z_n(Q_A) = \sum_{Q_A} \text{Tr}\left[\left(\rho_A^{(Q_A)}\right)^n\right],$$
$$\mathcal{S} = \sum_{Q_A} \mathcal{S}(Q_A) = -\sum_{Q_A} \text{Tr}\left[\rho_A^{(Q_A)} \ln \rho_A^{(Q_A)}\right]. \tag{4}$$

We note that the definition in Eq. (4) follows Refs. [56, 57], while in other works [58–60] each symmetry block is normalized by its trace before the resolved moments and entropies are calculated, rendering them measures of entanglement following a projection onto a symmetry sector. These normalized quantities may be straightforwardly derived from their non-normalized counterparts from Eq. (4) by relying on the fact that $Z_1(Q_A)$ – which is simply the charge distribution in subsystem $A$ – returns the required trace of each block. Specifically, the post-projection vNEE is given by

$$\sigma(Q_A) = \ln(Z_1(Q_A)) + \frac{\mathcal{S}(Q_A)}{Z_1(Q_A)}. \tag{5}$$

The resolved quantities in Eq. (4) do not quantify entanglement when used alone, but are more readily calculated, and can be also directly measured in experiments [57, 61–64].

Symmetry-resolved entanglement represents the internal structure that symmetry imposes on the entanglement spectrum, thus embodying the interplay between conservation laws and entanglement. It has been recently investigated analytically and numerically in various systems, in and out of equilibrium [65–87]. A common behavior in these systems is entanglement equipartition [58, 67–73], implying that, to leading order in $L$, the post-projection vNEE $\sigma(Q_A)$ is constant across symmetry sectors. The estimation of symmetry-resolved entanglement was shown to yield additional valuable insights, e.g. regarding topological phase transitions [68, 78, 81] and dissipation in noisy devices [84].

The main result of this work is the exact calculation of an unusual entanglement scaling for the steady state of a one-dimensional fermionic system out of equilibrium, using a generalization of the Fisher-Hartwig conjecture [88]. We study a model of a uniform tight-binding chain containing an arbitrary scattering region at its center, to which a DC bias voltage is applied at zero temperature, thereby leading to a current-carrying steady state. This steady state may be described by a pure eigenstate of the Hamiltonian, with different distributions for scattering states incoming from the left and from the right. This model is relevant to both electronic [89] and cold atom [90] systems.

We report that the subsystem entanglement entropy in this nonequilibrium steady state exhibits a volume-law scaling accompanied by an additive logarithmic correction. More precisely, we find that the vNEE of a subsystem of length $L$, when located far enough from the scattering region, obeys

$$\mathcal{S} \sim \left( -\int_{k_-}^{k_+} \frac{dk}{2\pi} \left[ |t(k)|^2 \ln\left(|t(k)|^2\right) + |r(k)|^2 \ln\left(|r(k)|^2\right) \right] \right) L + \mathcal{C}_{\log} \ln L + \mathcal{C}_{\text{const}} . \quad (6)$$

Here $|t(k)|^2$ and $|r(k)|^2$ are the transmission and reflection factors (respectively) of the scatterer for a plane wave with momentum $k$, $0 \leq k_- < k_+$ are the two different Fermi momenta for right- and left-propagating fermions, and $\mathcal{C}_{\log}, \mathcal{C}_{\text{const}}$ are constants. The extensive term of the vNEE is thus generated by momentum eigenstates within the bias voltage window, and the contribution of each state is equivalent to the classical mixture entropy of the corresponding transmission probability. The logarithmic term in Eq. (6) is a zero-temperature effect, that arises due to the sharp Fermi-Dirac jumps in the distribution of the plane waves. The coefficient $\mathcal{C}_{\log}$ is therefore a function of the Fermi momenta, for which we provide an exact expression as well. For $\mathcal{C}_{\text{const}}$ we obtain an approximate expression, justified under the assumption of a small bias voltage.

Furthermore, we expand the result in Eq. (6) by deriving the exact entanglement entropy asymptotics for the case where the chain hosts multiple interspersed scattering regions, finding a similar scaling law. This type of scaling has been previously encountered in rather specific instances, for example in certain excited states, in the ground state of 1D systems with long-range couplings [52, 91], and in the diffusive time-averaged state of a 1D interacting system [92, 93]. We show how this entanglement scaling can arise generically within a local 1D system in a time-independent state, generalizing previous results in particular cases [29–33].

Our calculation also produces analytical results for the symmetry-resolved entanglement, with respect to the total fermionic charge that is conserved in the system. While the post-projection vNEE exhibits equipartition to leading order as usual, we remarkably find that the first term breaking equipartition may grow with the size of the subsystem, and is antisymmetric in $Q_A - \langle Q_A \rangle$, the deviation of the charge in $A$ from its mean. To the best of our knowledge, this is the first system found to display such behavior.

The paper is organized as follows: In Sec. 2 we describe the general 1D nonequilibrium model, and derive expressions for the elements of its two-site correlation matrix. In Sec. 3 we review the definition of the generating function that captures all resolved and unresolved moments and entanglement entropies. By employing a generalization of the Fisher-Hartwig conjecture, we analytically derive the exact leading-order asymptotics of the generating function for a large subsystem, along with an approximate subleading correction. We use the result for the generating function to extract quantities relating to charge statistics and the asymptotic scaling of the vNEE in the nonequilibrium steady state. We also address charge-resolved entanglement, and determine the form of the first term breaking entanglement equipartition. In Sec. 4 we apply our general scheme to a specific model, where a single impurity site is responsible for the scattering. We use this concrete example to demonstrate central aspects of our results, and to show that the analytical calculation generally compares favorably with numerics. Sec. 5 details how the analytical results may be generalized to a subsystem on a chain containing multiple but distant scattering regions. In Sec. 6 we discuss our main conclusions and outline possible future directions. Appendix A is dedicated to further technical details of the derivation. Appendix B contains additional plots of the analytical results for the single impurity model, and points out the generic features that apply to the general model.

## 2 Model

The general model with which this paper is concerned is that of a long homogeneous one-dimensional fermionic tight-binding chain, which contains a finite inhomogeneous region near its middle that induces scattering, as schematically depicted in Fig. 1(a). The single-particle Hamiltonian of such a system can be written as

$$\mathcal{H} = -t \sum_{n=n_{\text{scat}}}^{N/2-1} (|n\rangle\langle n+1| + |-n\rangle\langle -n-1| + \text{h.c.}) + \mathcal{H}_{\text{scat}}, \tag{7}$$

where $t > 0$ is the hopping amplitude, $N+1$ is the number of chain sites ($N$ is assumed to be even), and $\mathcal{H}_{\text{scat}}$ is the term that will give rise to scattering. $\mathcal{H}_{\text{scat}}$ should involve only a few chain sites in the vicinity of $n = 0$, and possibly also hopping terms to additional side-attached sites in this small region. $n_{\text{scat}} \geq 0$ is an integer such that states $|n\rangle$ with $|n| > n_{\text{scat}}$ do not appear in the matrix representation of $\mathcal{H}_{\text{scat}}$. Subsystem $A$, for which we will estimate the entanglement measures, includes $L$ contiguous sites ($1 \ll L \ll N$) located to one side of the scattering region, and far away from it such that every site $n$ in $A$ obeys $|n| - n_{\text{scat}} \gg L$. For the sake of simplifying the notations we shall focus on the case where the subsystem is to the right of the scattering region, though this is of course an arbitrary choice.

The scattering matrix [94] related to this problem is a function of $k$, the lattice momentum,

$$S(k) = \begin{pmatrix} r_L(k) & t_R(k) \\ t_L(k) & r_R(k) \end{pmatrix}. \tag{8}$$

$S$ is unitary, and in particular $|r_L(k)|^2 + |t_R(k)|^2 = |r_R(k)|^2 + |t_L(k)|^2 = 1$. For the scattering amplitudes we use the convention that attributes a momentum $k > 0$ to a state incoming from the left and a momentum $-k < 0$ to a state incoming from the right, so that $S(k)$ is defined for $k > 0$. Scattering states constitute the single-particle energy eigenstates for energies $|E| < 2t$. We assume that the scattering potential does not support a half-bound state, i.e. a non-normalizable solution with energy $E = 2t$ or $E = -2t$, as is the generic case [95–97]. This condition entails that $t_R(k), t_L(k) \to 0$ as $k \to 0, \pi$ [98]. The addition of $\mathcal{H}_{\text{scat}}$ may, however, create bound states in the single-particle energy spectrum, with energies $|E| > 2t$.

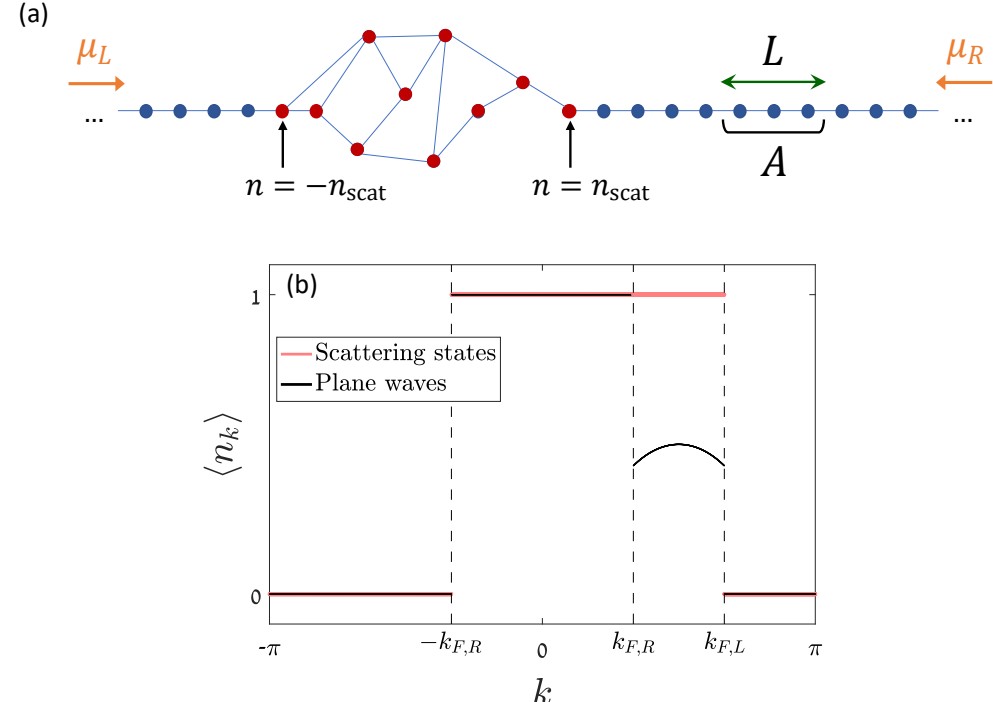

Figure 1: (a) Schematic illustration of the general lattice model under consideration in Secs. 2–3, with the single-particle Hamiltonian given in Eq. (7). Sites marked in blue belong to the unperturbed parts of the tight-binding chain, while sites marked in red belong to the scattering region, which terminates at the sites $n = \pm n_{\text{scat}}$. $\mu_L$ ($\mu_R$) designates the chemical potential for particles incoming from the left (right). $A$ denotes the subsystem of $L$ contiguous sites with respect to which the calculations of bipartite entanglement in Sec. 3 are performed. (b) Schematic plot of the distributions as functions of lattice momentum, for the scattering states (defined in Eqs. (9)–(10)) and for plane waves (in the region $n > n_{\text{scat}}$, to the right of the scattering region). The case shown is $k_{F,L} > k_{F,R}$.

If we denote by $|\psi_k^{(L)}\rangle$ the scattering state related to a wave incoming from the left with momentum $k > 0$, its form outside the scattering region will be

$$\langle n|\psi_k^{(L)}\rangle = \frac{1}{\sqrt{N}} \begin{cases} e^{ikn} + r_L(k)\,e^{-ikn} & n < -n_{\text{scat}}, \\ t_L(k)\,e^{ikn} & n > n_{\text{scat}}, \end{cases} \tag{9}$$

while for a wave incoming from the right with momentum $-k < 0$, we will have

$$\langle n|\psi_k^{(R)}\rangle = \frac{1}{\sqrt{N}} \begin{cases} t_R(k)\,e^{-ikn} & n < -n_{\text{scat}}, \\ e^{-ikn} + r_R(k)\,e^{ikn} & n > n_{\text{scat}}. \end{cases} \tag{10}$$

In the many-particle picture, the fermionic creation operator for a site located to the right of the scattering region, $n > n_{\text{scat}}$, can thus be expanded as

$$a_n^\dagger = \frac{1}{\sqrt{N}} \sum_{k>0} \left( e^{ikn} + r_R(k)^* e^{-ikn} \right) a_{k,R}^\dagger + \frac{1}{\sqrt{N}} \sum_{k>0} t_L(k)^* e^{-ikn} a_{k,L}^\dagger, \tag{11}$$

where $a_{k,R}^\dagger$ creates the scattering state $|\psi_k^{(R)}\rangle$, and $a_{k,L}^\dagger$ creates the scattering state $|\psi_k^{(L)}\rangle$. If $\mathcal{H}_{\text{scat}}$ indeed supports bound states, such states would appear as well in the superposition of

energy eigenstates that defines $|n\rangle$. We have nevertheless ignored these states in our writing $a_n^\dagger$ in terms of creation operators of energy eigenstates, due to the localized nature of such bound states, which makes their contribution to the entanglement exponentially small in the distance of subsystem $A$ from the scattering region.

Subject to an external constant bias voltage, such a system would arrive at a current-carrying steady state[1] where the Fermi momentum for waves incoming from the left, $k_{F,L}$, differs from the Fermi momentum for waves incoming from the right, $k_{F,R}$. At zero temperature this steady state is described by a pure many-body state,

$$|\Omega\rangle = \left(\prod_{0<k<k_{F,R}} a_{k,R}^\dagger\right)\left(\prod_{0<k<k_{F,L}} a_{k,L}^\dagger\right)|0\rangle, \tag{12}$$

where $|0\rangle$ is the vacuum state. Taking the limit $N \to \infty$, we replace sums over $k$ with appropriate integrals. The correlation between two sites $m, n > n_{\text{scat}}$ for this steady state is then given by

$$C_{mn} \equiv \langle a_m^\dagger a_n\rangle = \frac{1}{2\pi}\int_{-\pi}^{\pi} e^{-i(m-n)k}\tau(k)\,dk + \frac{1}{2\pi}\int_{-\pi}^{\pi} e^{-i(m+n)k}h(k)\,dk, \tag{13}$$

where

$$h(k) = \begin{cases} r_R(-k) & -k_{F,R} < k < 0, \\ r_R(k)^* & 0 < k < k_{F,R}, \\ 0 & \text{otherwise}, \end{cases} \tag{14}$$

and, in the case where $k_{F,R} < k_{F,L}$,

$$\tau(k) = \begin{cases} 1 & -k_{F,R} < k < k_{F,R}, \\ |t_L(k)|^2 & k_{F,R} < k < k_{F,L}, \\ 0 & \text{otherwise}. \end{cases} \tag{15}$$

If instead $k_{F,R} > k_{F,L}$, we obtain

$$\tau(k) = \begin{cases} 1 & -k_{F,R} < k < k_{F,L}, \\ |r_R(k)|^2 & k_{F,L} < k < k_{F,R}, \\ 0 & \text{otherwise}. \end{cases} \tag{16}$$

In the case where $m, n < -n_{\text{scat}}$, the result for $C_{mn}$ is similar up to the replacements $R \leftrightarrow L$, $\tau(k) \longrightarrow \tau(-k)$ and $h(k) \longrightarrow h(-k)$. As mentioned, subsystem $A$ is assumed to be located to the right of the scattering region, such that $n > n_{\text{scat}}$ for every site in $A$. For further convenience, we denote from now on $k_- = \min\{k_{F,R}, k_{F,L}\}$ and $k_+ = \max\{k_{F,R}, k_{F,L}\}$.

For each momentum $k$, the distribution $\langle n_k\rangle$ of the scattering states equals either 0 or 1 by the definition of the steady state. In contrast, the symbol $\tau(k)$ may be interpreted as the steady-state distribution of the unperturbed plane waves in the region $n > n_{\text{scat}}$; a plane wave state within the bias voltage window $k_- < k < k_+$ will generically be occupied with a fractional distribution factor $0 < \langle n_k\rangle < 1$. This distinction between the distributions of the

---

[1]A true steady state is reached only in the case of an infinite chain, $N \to \infty$, which is the limit examined here. One may consider a scenario where the system is prepared using a quench [99], i.e. by connecting at a specific point in time two separate leads with different chemical potentials via the scattering region. Once enough time has passed so that excitations that crossed from one lead to the other have traversed the finite subsystem, the particle fluxes incoming into and outgoing out of the subsystem become balanced, and the entanglement properties of the subsystem relax to time-independent values [36].

scattering states and the plane waves is illustrated in Fig. 1(b). As we shall see in Subsec. 3.4, the fractional occupation of the plane waves within the voltage window is the source of the extensive scaling of the subsystem entanglement.

We may conclude that the correlation matrix $C$ is a sum of a Toeplitz matrix, depending on the index difference $m-n$ (the integral in Eq. (13) containing $\tau(k)$), and a Hankel matrix, depending on the sum of indexes, $m+n$ (the integral in Eq. (13) containing $h(k)$). The Hankel term is negligible for $m, n \gg n_{\text{scat}}$ by virtue of the Riemann-Lebesgue lemma, and its decay is generically algebraic, $\frac{1}{2\pi} \int_{-\pi}^{\pi} e^{-i(m+n)k} h(k) \, dk = \mathcal{O}\left(\frac{1}{m+n}\right)$ [100]. We can thus write

$$C_{mn} \approx \frac{1}{2\pi} \int_{-\pi}^{\pi} e^{-i(m-n)k} \tau(k) \, dk, \quad \text{for } m, n \gg n_{\text{scat}}. \tag{17}$$

As we elaborate in Sec. 3, all analytical calculations included in this paper rely on the approximation in Eq. (17) of the two-site correlation matrix $C$. The use of this approximation is justified if the distance between subsystem $A$ and the scattering region is much larger than the length of $A$, as we further discuss in Subsec. 4.3. There we numerically demonstrate the algebraic decay of the contribution of the Hankel term (neglected within the approximation in Eq. (17)) to the Rényi moments, and also find that this contribution exhibits Friedel oscillations [101, 102].

## 3 Analytical asymptotics of the entanglement

In this section we present an analytical calculation of the Rényi moments and of the vNEE for subsystem $A$ with respect to its complement. Since in our model the total number of fermions in the lattice is conserved, the Rényi moments and the vNEE may also be resolved with respect to $Q_A = \sum_{m \in A} a_m^\dagger a_m$, the charge in subsystem $A$. The results we pursue are more conveniently calculated if we start by defining the following generating function [57, 58]:

$$Z_n(\alpha) = \text{Tr}\left[\rho_A^n e^{i\alpha Q_A}\right], \tag{18}$$

where $-\pi < \alpha < \pi$. The calculation of $Z_n(\alpha)$ actually encompasses all resolved and unresolved entropies and moments. Rényi moments are given by $Z_n = Z_n(\alpha = 0)$, from which the vNEE can be extracted through Eq. (3). Furthermore, the charge-resolved Rényi moment is simply a Fourier decomposition of the corresponding generating function [57],

$$Z_n(Q_A) = \int_{-\pi}^{\pi} \frac{d\alpha}{2\pi} Z_n(\alpha) e^{-i\alpha Q_A}. \tag{19}$$

In particular, $Z_1(\alpha)$ is the characteristic function of the charge distribution in $A$. The charge-resolved vNEE may be subsequently derived through

$$S(Q_A) = -\lim_{n \to 1} \partial_n Z_n(Q_A). \tag{20}$$

$Z_n(\alpha)$ can be written in terms of the eigenvalues $\{\nu_l\}$ of $2C_A - I_L$, where $I_L$ is the identity matrix of size $L$, and $C_A$ is the two-site correlation matrix (defined as $C$ in Eq. (13)) restricted to subsystem $A$. More concretely, we may write [57]

$$Z_n(\alpha) = \prod_{l=1}^{L} \left[\left(\frac{1+\nu_l}{2}\right)^n e^{i\alpha} + \left(\frac{1-\nu_l}{2}\right)^n\right], \tag{21}$$

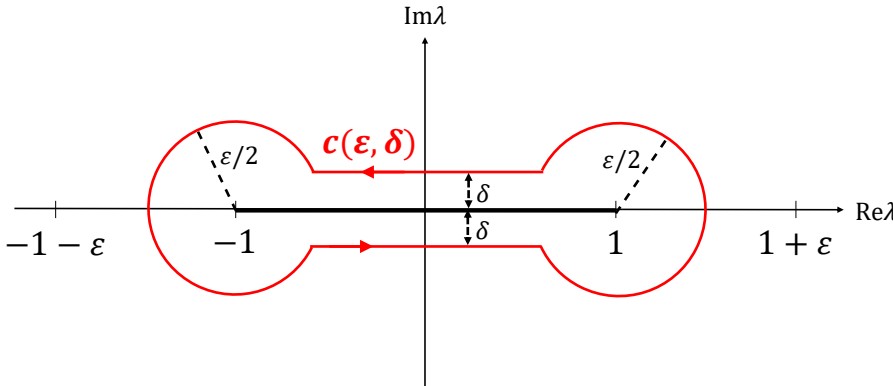

Figure 2: The integration contour used in Eq. (22).

which enables us to reformulate the calculation as a problem of contour integration in the complex plane [103]:

$$\ln Z_n(\alpha) = \lim_{\varepsilon,\delta \to 0^+} \frac{1}{2\pi i} \int_{c(\varepsilon,\delta)} e_n^{(\alpha)}(1+\varepsilon,\lambda) \frac{d}{d\lambda} \ln D_L(\lambda) d\lambda, \tag{22}$$

where $D_L(\lambda) = \det((\lambda+1)I_L - 2C_A)$ and $e_n^{(\alpha)}(x,\nu) = \ln\left[\left(\frac{x+\nu}{2}\right)^n e^{i\alpha} + \left(\frac{x-\nu}{2}\right)^n\right]$. The contour $c(\varepsilon,\delta)$ is defined such that it encloses the segment $[-1,1]$ of the real line (on which all the eigenvalues $\{\nu_l\}$ are located) while avoiding the singularities of $e_n^{(\alpha)}(1+\varepsilon,\lambda)$, as is depicted in Fig. 2.

### 3.1 Leading asymptotics of the generating function

The immediate consequence of Eq. (17) is that $D_L(\lambda)$ may be approximated as a Toeplitz determinant, meaning that $D_L(\lambda) = \det T_L(\lambda)$ where $T_L(\lambda)$ is a Toeplitz matrix. In particular, $(T_L)_{mn} = \phi_{m-n}$ where

$$\phi_l = \frac{1}{2\pi}\int_{-\pi}^{\pi} \phi(k)e^{-ilk}dk \; ; \; \phi(k) = \begin{cases} \lambda-1 & -k_{F,R} < k < k_-, \\ \lambda-\nu(k) & k_- < k < k_+, \\ \lambda+1 & \text{otherwise}. \end{cases} \tag{23}$$

Here we have denoted

$$\nu(k) = \begin{cases} |t_L(k)|^2 - |r_R(k)|^2 & k_{F,R} < k_{F,L}, \\ |r_R(k)|^2 - |t_L(k)|^2 & k_{F,R} > k_{F,L}, \end{cases} \tag{24}$$

a definition that may be more compactly packed into the form

$$\frac{1 \pm \nu(k)}{2} = |t_L(k)|^2 \quad \text{for } k_{F,L} = k_\pm \ (\text{resp. } k_{F,R} \lessgtr k_{F,L}). \tag{25}$$

Importantly, $|\nu(k)| \leq 1$.

The symbol $\phi(k)$ as defined in Eq. (23) cannot be cast in the Fisher-Hartwig form [88], contrary to what is required by the well-known (and proven) formulae and theorems of which we are aware [88,104,105] concerning the asymptotics of Toeplitz determinants. A generalized asymptotic formula for the determinant of a Toeplitz matrix generated by a piecewise-continuous symbol was conjectured in Refs. [106,107]. According to this formula, for $L \gg 1$

we have

$$\ln D_L(\lambda) = \frac{L}{2\pi}\int_{-\pi}^{\pi} dk \ln\phi(k) + \frac{\ln L}{4\pi^2}\left[\left(\ln\frac{\lambda-1}{\lambda-\nu(k_-)}\right)^2 + \left(\ln\frac{\lambda-\nu(k_+)}{\lambda+1}\right)^2 + \left(\ln\frac{\lambda+1}{\lambda-1}\right)^2\right] + \dots,$$

(26)

where the ellipses represent terms of lower order in $L$.

Plugging the asymptotic form in Eq. (26) into the integral expression in Eq. (22), we obtain

$$\ln Z_n(\alpha) \sim \frac{L}{2\pi}\left[i\alpha\left(k_- + k_{F,R}\right) + \int_{k_-}^{k_+} e_n^{(\alpha)}(1,\nu(k))\,dk\right]$$
$$+ \ln L\left[Q_n(\nu(k_-),\alpha) + Q_n(-\nu(k_+),-\alpha) + \frac{1}{12}\left(\frac{1}{n}-n\right) - \frac{\alpha^2}{4\pi^2 n}\right]$$
$$\equiv \mathcal{I}_{\text{lin}}(n,\alpha)L + \mathcal{I}_{\text{log}}(n,\alpha)\ln L,$$

(27)

where we have defined

$$Q_n(\nu,\alpha) = \frac{1}{2\pi^2}\int_\nu^1 \ln\left|\frac{x-1}{x-\nu}\right|\frac{d}{dx}e_n^{(\alpha)}(1,x)\,dx.$$

(28)

The term $\mathcal{I}_{\text{lin}}(n,\alpha)$ is derived in a straightforward manner, while a detailed derivation of the term $\mathcal{I}_{\text{log}}(n,\alpha)$ appears in Appendix A.1.

The linear term in $L$ appearing in Eq. (27) counts the filled momentum (plane wave) states. The states with $-k_{F,R} < k < k_-$ are all filled with probability 1, while if $k_{F,R} < k_{F,L}$, the states with $k_- < k < k_+$ are filled with probability $|t_L(k)|^2$ and empty with probability $1 - |t_L(k)|^2$ (or vice versa in the case where $k_{F,L} < k_{F,R}$). This distribution is schematically presented in Fig. 1(b). Since any interval $\delta k$ includes $(L/2\pi)\delta k$ states, $\exp(\mathcal{I}_{\text{lin}}(n,\alpha)L)$ is simply the product of the moments that arise from the individual filled states. The term $\exp(\mathcal{I}_{\text{lin}}(n,\alpha)L)$ can thus be interpreted as a generalization of the generating function for the full counting statistics in the case of a transmission factor that is constant in $k$, cf. Eq. (20) in Ref. [99] (the Levitov-Lesovik formula).

The logarithmic term in $L$ appearing in Eq. (27) is a result of the Fermi discontinuities at $k = k_\pm$ and $k = -k_{F,R}$, featured in Fig. 1(b). In particular, since $Q_n(1,\alpha) = 0$ by definition, the contribution from $k = k_\pm$ vanishes when $\nu(k_\pm) \to \mp 1$ (respectively), in accordance with the disappearance of the respective jump discontinuity of the symbol $\phi(k)$. That is also the case when the bias voltage is larger than the bandwidth, such that $k_{F,L} = 0$ and $k_{F,R} = \pi$ or vice versa: since $|r_R(k)| \to 1$ as $k \to 0, \pi$, the symbol $\phi(k)$ is continuous at any value of $k$, and thus the logarithmic term vanishes from Eq. (27).

## 3.2 Subleading asymptotics of the generating function

In order to incorporate further subleading terms in the analytical asymptotics of $\ln Z_n(\alpha)$, we now approximate the symbol $\phi(k)$ in Eq. (23) to be piecewise-constant, such that it will fit the Fisher-Hartwig form [88]. The underlying assumption is that under a small bias such that $\Delta k \equiv |k_{F,L} - k_{F,R}| \ll 1$ it would be permissible to use the approximation $\nu(k) \approx \nu(k_0)$ for $k_- < k < k_+$, where $k_0 \equiv \frac{1}{2}(k_{F,R} + k_{F,L})$. This requires the transmission and reflection factors

to change slowly near $k = k_0$. For this purpose we define the following approximate symbol:

$$\tilde{\phi}(k) = \begin{cases} \lambda - 1 & -k_{F,R} < k < k_-, \\ \lambda - v_0 & k_- < k < k_+, \\ \lambda + 1 & \text{otherwise}, \end{cases} \tag{29}$$

where we denoted $v_0 = v(k_0)$.

Let us denote by $\tilde{D}_L(\lambda)$ the determinant of the Toeplitz matrix generated by the symbol in Eq. (29). Using the Fisher-Hartwig formula [88, 103] for the asymptotics of $\tilde{D}_L(\lambda)$ and substituting it into the integral expression (22) for the generating function will yield for $L \gg 1$ an expression of the form

$$\ln Z_n(\alpha) \approx \tilde{\mathcal{I}}_{\text{lin}}(n, \alpha) L + \tilde{\mathcal{I}}_{\text{log}}(n, \alpha) \ln L + \tilde{\mathcal{I}}_{\text{const}}(n, \alpha). \tag{30}$$

Note that while the asymptotics of $\ln Z_n(\alpha)$ in Eq. (27) was estimated up to a linear term and a logarithmic term in $L$ without an approximation of the Toeplitz symbol, here the approximation $\tilde{\phi}(k)$ yields an additional term which is independent of $L$. Compared to the exact expressions for the linear and logarithmic terms, the error of the terms obtained from the approximate symbol scale as

$$\begin{aligned} \mathcal{I}_{\text{lin}}(n, \alpha) - \tilde{\mathcal{I}}_{\text{lin}}(n, \alpha) &\sim (\Delta k)^3, \\ \mathcal{I}_{\text{log}}(n, \alpha) - \tilde{\mathcal{I}}_{\text{log}}(n, \alpha) &\sim (\Delta k)^2 \ln \Delta k, \end{aligned} \tag{31}$$

as $\Delta k \to 0$.

Relying on the Fisher-Hartwig formula, we obtain that

$$\begin{aligned} \tilde{\mathcal{I}}_{\text{const}}(n, \alpha) = {}& \ln \left| \frac{2 \sin \frac{k_- + k_{F,R}}{2} \sin \frac{1}{2} \Delta k}{\sin \frac{k_+ + k_{F,R}}{2}} \right| Q_n(v_0, \alpha) \\ &+ \ln \left| \frac{2 \sin \frac{k_+ + k_{F,R}}{2} \sin \frac{1}{2} \Delta k}{\sin \frac{k_- + k_{F,R}}{2}} \right| Q_n(-v_0, -\alpha) \\ &+ \ln \left| \frac{2 \sin k_{F,R} \sin k_0}{\sin \frac{1}{2} \Delta k} \right| \left[ \frac{1}{12} \left( \frac{1}{n} - n \right) - \frac{\alpha^2}{4\pi^2 n} \right] \\ &+ \Upsilon_n(v_0, \alpha) + \Upsilon_n(-v_0, -\alpha) + \Upsilon_n(-1, \alpha), \end{aligned} \tag{32}$$

where we have defined

$$\Upsilon_n(v, \alpha) = \frac{1}{2\pi i} \int_v^1 \ln \frac{\Gamma\left(\frac{1}{2} + \frac{1}{2\pi i} \ln\left(\frac{1-x}{x-v}\right)\right)}{\Gamma\left(\frac{1}{2} - \frac{1}{2\pi i} \ln\left(\frac{1-x}{x-v}\right)\right)} \frac{d}{dx} e_n^{(\alpha)}(1, x) \, dx. \tag{33}$$

We detail the calculation of $\tilde{\mathcal{I}}_{\text{const}}(n, \alpha)$ in Appendix A.2. In total, the approximate asymptotic expression for the generating function is

$$\ln Z_n(\alpha) \approx \mathcal{I}_{\text{lin}}(n, \alpha) L + \mathcal{I}_{\text{log}}(n, \alpha) \ln L + \tilde{\mathcal{I}}_{\text{const}}(n, \alpha). \tag{34}$$

Note that, generically, the terms in $\mathcal{I}_{\text{log}}(n, \alpha)$ and $\tilde{\mathcal{I}}_{\text{const}}(n, \alpha)$ which stem from the partial transmission effects for $k_- < k < k_+$ do not vanish when we fix $L$ and simply take the limit $\Delta k \to 0$; we must first take $v(k) \to \pm 1$ in the interval $k_- < k < k_+$ in order for them to vanish. This, however, is in compliance with the fact that the problem is examined at the limit of large $L$, which thus constitutes the largest length scale of the problem (other than the

length scales that are assumed infinite, i.e. the length of the full chain and the distance of $A$ from the scattering region). The limits $L \to \infty$ and $\Delta k \to 0$ do not commute; in other words, our expressions assume $\Delta k \gg 1/L$, and hence taking $\Delta k \to 0$ in them does not eliminate the contributions coming from the partial transmission within the voltage window $\Delta k$.

A more problematic feature of the analytical result is that for any $n$, the function $Q_n(v, \alpha)$ (which appears in the expressions for both $\mathcal{I}_{\log}(n, \alpha)$ and $\tilde{\mathcal{I}}_{\text{const}}(n, \alpha)$) is singular at $v = 0$ and $\alpha = \pm\pi$. Moreover, it may be shown that

$$\text{Re}\, Q_n(0, \alpha) \sim \frac{1}{4\pi^2} \ln^2\left(\frac{\pi \mp \alpha}{2n}\right), \quad \text{as } \alpha \to \pm\pi. \tag{35}$$

This deems the analytical expression for $Z_n(\alpha)$ to be a non-integrable function of $\alpha$ over $[-\pi, \pi]$ if either $v(k_+) = 0$ or $v(k_-) = 0$. Numerical results do not exhibit this kind of divergence, and so this property of the analytical result does not capture the true behavior of the generating function. The singularity of the function $e_n^{(\alpha)}(1, v)$ (defined right after Eq. (22)) at $v = 0, \alpha = \pm\pi$ is what brings about this difficulty, as small shifts of $v(k_\pm)$ become crucial when either one is near $v = 0$.

We note that our previous work [68], which had discussed a situation where the Toeplitz symbol may indeed be cast in the Fisher-Hartwig form, established that corrections to the approximation of $\ln Z_n(\alpha)$ using the Fisher-Hartwig formula decay less rapidly with $L$ as $\alpha$ nears $\pm\pi$. At $\alpha = \pm\pi$ these corrections eventually become as important as the terms in Eq. (30), causing a considerable deviation from exact numerical results if one does not include the corrections [67, 68, 70]. Although there is no known expression for these corrections when the Toeplitz symbol does not fit the Fisher-Hartwig form, we expect them to eliminate the divergence at $\alpha = \pm\pi$ observed in this case. While the divergence of the generating function prevents us from obtaining analytical results for charge-resolved quantities in cases where either $v(k_+)$ or $v(k_-)$ exactly vanish, we have found that it has little effect whenever $v(k_\pm)$ are finite, even when they are small, as demonstrated in Subsec. 4.2.

We may also estimate the deviation of the generating function for the nonequilibrium steady state from that of the ground state in the equilibrium case. In Refs. [67, 68] it was shown that the leading-order asymptotics of the generating function for the ground state of a homogeneous tight-binding chain filled up to $k = \pm k_0$ is given by[2]

$$\ln Z_n^{\text{eq}}(\alpha) \approx i \frac{k_0 \alpha}{\pi} L + \left[\frac{1}{6}\left(\frac{1}{n} - n\right) - \frac{\alpha^2}{2\pi^2 n}\right] \ln|2L \sin k_0| + 2\Upsilon_n(-1, \alpha). \tag{36}$$

Subtracting this from the nonequilibrium result, we obtain

$$
\begin{aligned}
\ln \frac{Z_n(\alpha)}{Z_n^{\text{eq}}(\alpha)} \approx \ & \frac{L}{2\pi}\left[ i\alpha\left(k_- - k_{F,L}\right) + \int_{k_-}^{k_+} e_n^{(\alpha)}(1, v(k))\, dk \right] \\
& + \ln L\left[ Q_n(v(k_-), \alpha) + Q_n(-v(k_+), -\alpha) - \frac{1}{12}\left(\frac{1}{n} - n\right) + \frac{\alpha^2}{4\pi^2 n} \right] \\
& + \ln\left| \frac{2 \sin \frac{k_- + k_{F,R}}{2} \sin \frac{1}{2}\Delta k}{\sin \frac{k_+ + k_{F,R}}{2}} \right| Q_n(v_0, \alpha) \\
& + \ln\left| \frac{2 \sin \frac{k_+ + k_{F,R}}{2} \sin \frac{1}{2}\Delta k}{\sin \frac{k_- + k_{F,R}}{2}} \right| Q_n(-v_0, -\alpha)
\end{aligned}
$$

---

[2]This is equal to the expression obtained for the nonequilibrium steady state with a nonzero bias voltage, but in the absence of scattering, i.e. assuming $t_L(k) = 1$ for all $k$.

$$- \ln \left| \frac{2 \sin \frac{1}{2} \Delta k \sin k_0}{\sin k_{F,R}} \right| \left[ \frac{1}{12} \left( \frac{1}{n} - n \right) - \frac{\alpha^2}{4\pi^2 n} \right]$$
$$+ \Upsilon_n (\nu_0, \alpha) + \Upsilon_n (-\nu_0, -\alpha) - \Upsilon_n (-1, \alpha). \tag{37}$$

### 3.3 Charge statistics

An important special case of the symmetry-resolved Rényi moments is that of $n = 1$, since $Z_1(Q_A)$ constitutes the charge distribution in subsystem $A$, and an expansion of $\ln Z_1(\alpha)$ in powers of $\alpha$ gives its moments. Indeed, we may write

$$\ln \frac{Z_1(\alpha)}{Z_1^{\text{eq}}(\alpha)} = i \left( \langle Q_A \rangle - \langle Q_A \rangle_{\text{eq}} \right) \alpha - \frac{1}{2} \left[ (\Delta Q_A)^2 - (\Delta Q_A)_{\text{eq}}^2 \right] \alpha^2 + \mathcal{O}(\alpha^3), \tag{38}$$

where

$$\langle Q_A \rangle - \langle Q_A \rangle_{\text{eq}} = -\frac{1}{2\pi} \left[ \int_{k_{F,R}}^{k_{F,L}} |r_R(k)|^2 \, dk \right] L \tag{39}$$

is the shift in the mean charge, and

$$(\Delta Q_A)^2 - (\Delta Q_A)_{\text{eq}}^2 = \frac{1}{2\pi} \left( \int_{k_-}^{k_+} |t_L(k) r_R(k)|^2 \, dk \right) L$$
$$- \frac{1}{2\pi^2} \left( 1 - |r_R(k_{F,R})|^4 - |t_L(k_{F,L})|^4 \right) \ln L$$
$$+ \frac{|r_R(k_0)|^2}{\pi^2} \ln \left| \frac{\sin k_{F,R}}{\sin k_0} \right|$$
$$- \frac{|t_L(k_0) r_R(k_0)|^2}{\pi^2} \left( 1 + \gamma_E + \ln \left| 2 \sin \frac{\Delta k}{2} \right| \right) \tag{40}$$

is the shift in the charge variance. Here $\gamma_E \approx 0.577$ is the Euler-Mascheroni constant [108]. A derivation of Eqs. (39) and (40) appears in Appendix A.3. Let us note that the equilibrium values of the mean and variance of the charge in subsystem $A$ are [68]

$$\langle Q_A \rangle_{\text{eq}} = \frac{k_0}{\pi} L \ , \ (\Delta Q_A)_{\text{eq}}^2 = \frac{\ln |2L \sin k_0| + 1 + \gamma_E}{\pi^2}. \tag{41}$$

Analogously, we may define a generalized quantity $\langle Q_A \rangle_n = -i \partial_\alpha \ln Z_n(\alpha)|_{\alpha=0}$, designating the mean of the "charge distribution" whose characteristic function is $Z_n(\alpha)$. By taking the derivative of Eq. (34), this generalized mean charge is found to be

$$\langle Q_A \rangle_n = \left[ k_- + k_{F,R} + \int_{k_-}^{k_+} \frac{(1 + \nu(k))^n}{(1 + \nu(k))^n + (1 - \nu(k))^n} \, dk \right] \frac{L}{2\pi}$$
$$+ \left[ \int_{\nu(k_-)}^{1} \ln \left| \frac{x - 1}{x - \nu(k_-)} \right| \mathfrak{g}_n(x) \, dx - \int_{-\nu(k_+)}^{1} \ln \left| \frac{x - 1}{x + \nu(k_+)} \right| \mathfrak{g}_n(x) \, dx \right] \frac{\ln L}{\pi^2}$$
$$+ \frac{1}{\pi^2} \ln \left| \frac{2 \sin \frac{k_- + k_{F,R}}{2} \sin \frac{1}{2} \Delta k}{\sin \frac{k_+ + k_{F,R}}{2}} \right| \int_{\nu_0}^{1} \ln \left| \frac{x - 1}{x - \nu_0} \right| \mathfrak{g}_n(x) \, dx$$

$$
-\frac{1}{\pi^2}\ln\left|\frac{2\sin\frac{k_+ + k_{F,R}}{2}\sin\frac{1}{2}\Delta k}{\sin\frac{k_- + k_{F,R}}{2}}\right|\int_{-v_0}^{1}\ln\left|\frac{x-1}{x+v_0}\right|\mathfrak{g}_n(x)\,dx
$$

$$
+\frac{1}{\pi i}\int_{v_0}^{1}\ln\frac{\Gamma\left(\frac{1}{2}+\frac{1}{2\pi i}\ln\left(\frac{1-x}{x-v_0}\right)\right)}{\Gamma\left(\frac{1}{2}-\frac{1}{2\pi i}\ln\left(\frac{1-x}{x-v_0}\right)\right)}\mathfrak{g}_n(x)\,dx
$$

$$
-\frac{1}{\pi i}\int_{-v_0}^{1}\ln\frac{\Gamma\left(\frac{1}{2}+\frac{1}{2\pi i}\ln\left(\frac{1-x}{x+v_0}\right)\right)}{\Gamma\left(\frac{1}{2}-\frac{1}{2\pi i}\ln\left(\frac{1-x}{x+v_0}\right)\right)}\mathfrak{g}_n(x)\,dx\,, \tag{42}
$$

where we have denoted $\mathfrak{g}_n(x)=\frac{n(1-x^2)^{n-1}}{[(1+x)^n+(1-x)^n]^2}$. In similar fashion, one can obtain an analytical expression for the corresponding variance by calculating $(\Delta Q_A)_n^2=-\partial_\alpha^2\ln Z_n(\alpha)|_{\alpha=0}$, and in particular find that generically it scales linearly with $L$.

### 3.4 Unresolved entanglement

By setting $\alpha=0$ in the generating function from Eq. (34), we obtain analytical expressions for the unresolved entanglement measures we wished to estimate. Rényi moments and entropies are directly accessible in this manner, while the vNEE is extracted through its relation to the Rényi moments, per Eq. (3). In order to conveniently present the resultant asymptotics for the vNEE, we define the following functions:

$$
q(p)=\frac{1}{8}-\frac{p}{24}-\frac{1}{2\pi^2}\int_0^1\frac{dx}{x}\left\{\frac{(1+px)\ln(1+px)+(x+p)\ln(x+p)}{1+x}-p\ln p\right\},
$$

$$
v(p)=\kappa_0-\frac{1}{2\pi^2}\int_0^1\frac{dx}{x}\left\{\frac{(1+px)\ln(1+px)+(x+p)\ln(x+p)+(1-p)x\ln x}{1+x}-p\ln p\right\}
$$

$$
\times\int_0^\infty\left[\frac{\cos\left(\frac{\ln x}{2\pi}z\right)}{2\sinh\left(\frac{z}{2}\right)}-\frac{e^{-z}}{z}\right]dz\,, \tag{43}
$$

where we have introduced the constant $\kappa_0=\int_0^\infty\left[\frac{1}{z^2\sinh\left(\frac{z}{2}\right)}-\frac{1}{2z\sinh^2\left(\frac{z}{2}\right)}-\frac{e^{-z}}{12z}\right]dz\approx 0.1399$.

We then have for the vNEE the following result:

$$
\mathcal{S}\sim\mathcal{C}_{\text{lin}}L+\mathcal{C}_{\text{log}}\ln L+\mathcal{C}_{\text{const}}\,, \tag{44}
$$

where $\mathcal{C}_{\text{lin}}$ and $\mathcal{C}_{\text{log}}$ are both exact and are given by

$$
\mathcal{C}_{\text{lin}}=-\frac{1}{2\pi}\int_{k_-}^{k_+}\left[|t_L(k)|^2\ln\left(|t_L(k)|^2\right)+|r_R(k)|^2\ln\left(|r_R(k)|^2\right)\right]dk \tag{45}
$$

and

$$
\mathcal{C}_{\text{log}}=\frac{1}{6}+q\left(\left|t_L\left(k_{F,R}\right)\right|^2\right)+q\left(\left|r_R\left(k_{F,L}\right)\right|^2\right)\,, \tag{46}
$$

and $\mathcal{C}_{\text{const}}$ is the approximate constant correction (valid, as before, when $\Delta k$ is small enough),

$$
\begin{aligned}
\mathcal{C}_{\text{const}} = {} & \ln\left|\frac{2\sin k_{F,R}\sin\frac{1}{2}\Delta k}{\sin k_0}\right| q\left(|t_L(k_0)|^2\right) + \ln\left|\frac{2\sin k_0\sin\frac{1}{2}\Delta k}{\sin k_{F,R}}\right| q\left(|r_R(k_0)|^2\right) \\
& + \frac{1}{6}\ln\left|\frac{2\sin k_{F,R}\sin k_0}{\sin\frac{1}{2}\Delta k}\right| + \upsilon\left(|t_L(k_0)|^2\right) + \upsilon\left(|r_R(k_0)|^2\right) + \upsilon(0).
\end{aligned}
\tag{47}
$$

This is derived with further details in Appendix A.4. Eq. (44) was already highlighted in Sec. 1 (where it is featured as Eq. (6)) as a central result of this work.

The form of $\mathcal{C}_{\text{lin}}$ in Eq. (45) is especially illuminating. The extensive term of the entanglement entropy arises from the integration of a classical mixture entropy with respect to the reflection and transmission probabilities. It also highlights the two crucial ingredients that produce the linear leading term: the nonequilibrium setting brought about by the bias voltage, which ensures that $k_+ \neq k_-$ and therefore that the integral does not trivially vanish; and a scattering potential that generates imperfect transmission, seeing that a unity transmission factor will cause the integrand in Eq. (45) to vanish for all $k$. Both conditions must apply in order for the leading term of $\mathcal{S}$ to be extensive (for related treatments of particular time-dependent impurity setups, see Refs. [29, 32]).

## 3.5 Charge-resolved entanglement

An exact computation of the charge-resolved Rényi moments based on the analytical asymptotics of the generating function in Eq. (34) requires carrying out the integration in Eq. (19), which cannot itself be performed analytically. For $L \gg 1$, a useful approximation is obtained by expanding $\ln Z_n(\alpha)$ in powers of $\alpha$ up to second order and replacing the integration limits in Eq. (19) by $\pm\infty$. This leads to an approximate Gaussian form of the charge-resolved $n$th Rényi moment:

$$
Z_n(Q_A) \approx \frac{Z_n}{\sqrt{2\pi(\Delta Q_A)_n^2}}\exp\left[-\frac{(Q_A - \langle Q_A\rangle_n)^2}{2(\Delta Q_A)_n^2}\right].
\tag{48}
$$

The above approximation holds since for large $L$, $(\Delta Q_A)_n^2$ scales linearly with $L$ (as mentioned in Subsec. 3.3), and consequently $Z_n(\alpha)$ decays rapidly away from $\alpha = 0$ (this is analogous to the central limit theorem).

The Gaussian approximation allows us to analytically examine the question of entanglement equipartition. By plugging Eq. (48) into Eq. (5), we obtain the following expression for the vNEE after a projective charge measurement:

$$
\begin{aligned}
\sigma(Q_A) \approx {} & \mathcal{S} - \frac{1}{2}\ln\left(2\pi(\Delta Q_A)^2\right) - \frac{\partial_n\langle Q_A\rangle_n|_{n=1}}{(\Delta Q_A)^2}(Q_A - \langle Q_A\rangle) \\
& - \frac{1}{2}\left(\frac{Q_A - \langle Q_A\rangle}{\Delta Q_A}\right)^2 + \frac{\partial_n(\Delta Q_A)_n|_{n=1}}{\Delta Q_A}\left(1 - \left(\frac{Q_A - \langle Q_A\rangle}{\Delta Q_A}\right)^2\right).
\end{aligned}
\tag{49}
$$

Relying on Eqs. (40) and (42), we note that both $(\Delta Q_A)^2$ and $\partial_n\langle Q_A\rangle_n|_{n=1}$ scale linearly with $L$ to leading order. In particular, the approximation in Eq. (49) is expected to be valid for values of $Q_A$ obeying $Q_A - \langle Q_A\rangle = \mathcal{O}\left(\sqrt{L}\right)$. Eq. (49) entails that, to leading (linear in $L$) order, entanglement is spread equally among charge sectors with $\sigma(Q_A) \approx \mathcal{S}$, but also that this equipartition may be broken by a term up to order $\mathcal{O}\left(\sqrt{L}\right)$. To the best of our knowledge, this is the first calculation of symmetry-resolved entanglement entropy showing a term breaking equipartition that grows with the size of the subsystem in question [67–73].

The fact that the first term breaking entanglement equipartition is odd with respect to $Q_A - \langle Q_A \rangle$ is noteworthy as well. Previous works that explicitly calculated the first equipartition-breaking term in different equilibrium and nonequilibrium models have always found it to be an even function of the deviation from the mean charge [67–73]. For $L \gg 1$ this odd term is given by

$$-\frac{\partial_n \langle Q_A \rangle_n |_{n=1}}{(\Delta Q_A)^2}(Q_A - \langle Q_A \rangle) \approx \frac{\int_{k_-}^{k_+}\left(1 - \nu(k)^2\right)\ln\frac{1-\nu(k)}{1+\nu(k)}dk}{\int_{k_-}^{k_+}\left(1 - \nu(k)^2\right)dk}(Q_A - \langle Q_A \rangle)\,, \tag{50}$$

which suggests that, for a small nonzero bias voltage, the post-measurement vNEEs of two charge sectors $Q_A = q_1$ and $Q_A = q_2$ approximately differ by

$$\sigma(q_1) - \sigma(q_2) \approx \ln\left(\frac{1 - \nu_0}{1 + \nu_0}\right)(q_1 - q_2)\,. \tag{51}$$

In the cases of either zero bias voltage ($\Delta k = 0$) or perfect transmission or reflection ($t_L(k) = 1$ or $t_L(k) = 0$, respectively, for all $k$), we have $\partial_n \langle Q_A \rangle_n = 0$, so the equipartition-breaking term that displays both these novel features vanishes. We stress that these features are unique even with respect to the case studied in Ref. [73], where entanglement equipartition is examined for a nonequilibrium steady state created following a global quench. The main distinction between the steady state in Ref. [73] and the steady state we investigated here is that the latter is a state with a net current that is partially transmitted by the scatterer.

## 4 The single impurity model

We consider a concrete example of the general model discussed above, by setting an on-site energy cost for the middle site of the chain. The single-particle Hamiltonian in Eq. (7) becomes

$$\mathcal{H} = -t\sum_{n=-\infty}^{\infty}(|n\rangle\langle n+1| + |n+1\rangle\langle n|) + \epsilon_0|0\rangle\langle 0|\,, \tag{52}$$

where $\epsilon_0 \in \mathbb{R}$. The scattering states which constitute solutions for the single-particle problem provide the following transmission and reflection coefficients:

$$\left|t_{R,L}(k)\right|^2 = \frac{\sin^2 k}{\sin^2 k + (\epsilon_0/2t)^2} \quad,\quad \left|r_{R,L}(k)\right|^2 = \frac{(\epsilon_0/2t)^2}{\sin^2 k + (\epsilon_0/2t)^2}\,. \tag{53}$$

There is also a bound state with energy $E > 2t$ ($E < -2t$) for $\epsilon_0 > 0$ ($\epsilon_0 < 0$); however, as noted above, its contribution is negligible in the limit considered. The generating function $Z_n(\alpha)$ thus depends on the parameters $k_{\pm}$ and $\epsilon_0/t$, along with the more explicit dependence on $\alpha$ and $n$. Appendix B illustrates how the coefficients that define the analytical asymptotic expression for $\ln Z_n(\alpha)$ in Eq. (34) vary with these parameters.

In this section we first focus on measures of entanglement extracted from the generating function for the single impurity model, with a comparison of our analytical results to numerics (Subsecs. 4.1 and 4.2). We then use numerics for this model to discuss more generally the accuracy of the calculation of the generating function (Subsec. 4.3).

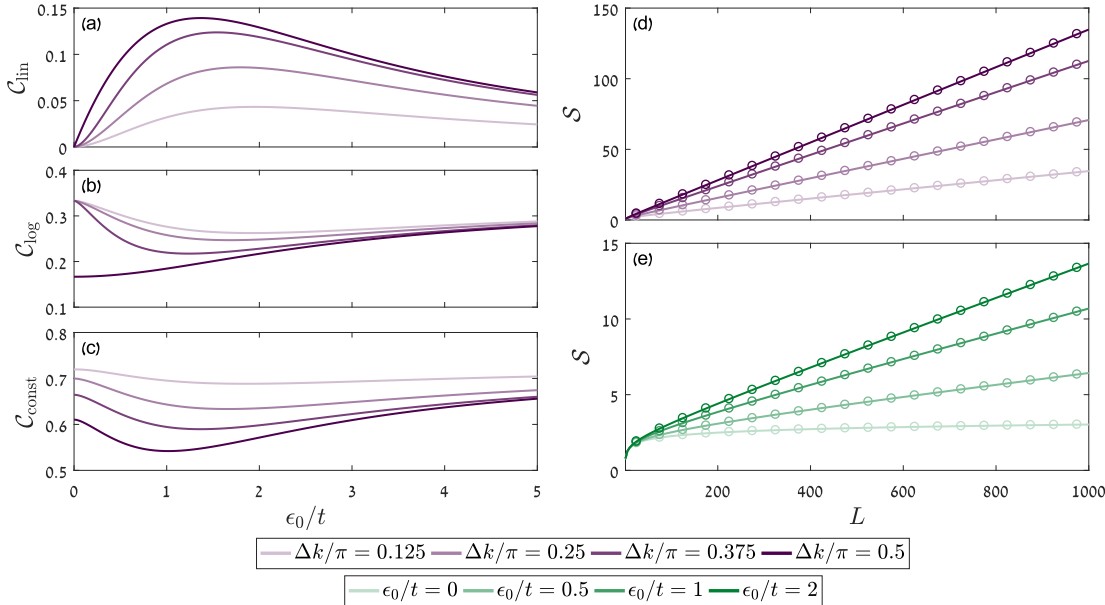

Figure 3: The single impurity model: Unresolved vNEE. (a)–(c) Asymptotic scaling coefficients, as defined in Eqs. (45)–(47), as functions of $\epsilon_0/t$ for various fixed values of $\Delta k = k_{F,L} - k_{F,R}$, with $k_{F,R} = \pi/2$. (d)–(e) The vNEE $\mathcal{S}$ as a function of the length $L$ of subsystem $A$, calculated analytically (lines) using Eq. (44), and numerically (circles) using Eqs. (3), (17) and (21). In (d) we show results for various fixed values of $\Delta k = k_{F,L} - k_{F,R}$, with $\epsilon_0 = t$ and $k_{F,R} = \pi/2$; in (e) we show results for various fixed values of $\epsilon_0/t$, with $k_{F,R} = \pi/2$ and $k_{F,L} - k_{F,R} = 0.1$.

## 4.1 Unresolved von-Neumann entanglement entropy

As already stressed, the entanglement between subsystem $A$ and its complement is most rigorously quantified by the vNEE. In Fig. 3 we plot the dependence on the model parameters of the asymptotic scaling coefficients of the vNEE from Eq. (44). It can be seen that the coefficients are nonmonotonic in $\epsilon_0/t$. Notably, the integral form of $\mathcal{C}_{\text{lin}}$ in Eq. (45) suggests that the largest contribution to the leading extensive term of $\mathcal{S}$ comes form momentum states where $|t_L(k)|^2 \approx |r_R(k)|^2$; this, in turn, implies that $\mathcal{C}_{\text{lin}}$ should peak at a value of $\epsilon_0/t$ such that $|t_L(k_0)|^2 \approx |r_R(k_0)|^2$, as is evident in Fig. 3(a).

Another noteworthy detail is that Fig. 3(b) exemplifies the non-continuous nature of the asymptotic scaling coefficients that depend on the Fermi discontinuities. Indeed, in a homogeneous chain ($\epsilon_0 = 0$) we have $\mathcal{C}_{\text{log}} = 1/3$ for any $0 < k_0 < \pi$, but Fig. 3(b) shows that by fixing $k_{F,L} = \pi$ first and taking the limit $\epsilon_0/t \to 0$ later, we obtain $\mathcal{C}_{\text{log}} \to 1/6$. This is because for any $\epsilon_0 \neq 0$ there is no Fermi discontinuity at $k_{F,L} = \pi$, but one is created at $\epsilon_0 = 0$. One must therefore be careful when taking limits that either create or destroy jumps in the distribution.

To corroborate these analytical results, the vNEE was also extracted through Eq. (3) from a numerical calculation of the Rényi moments $Z_n$. The latter is performed using the exact expression for the generating function in Eq. (21), where the restricted correlation matrix $C_A$ is approximated according to Eq. (17), so that effects of a finite distance between subsystem $A$ and the impurity site are neglected. Figs. 3(d)–(e) feature a comparison of the numerical result for the vNEE with our analytical calculation, confirming good agreement between them. As attested by Fig. 3(d), this is true even for a bias voltage that is not relatively small, such that $\Delta k \approx 1$. Recall that the assumption $\Delta k \ll 1$ was required only for justifying the approximation of $\mathcal{C}_{\text{const}}$, while both leading terms of the asymptotics are exact regardless of it.

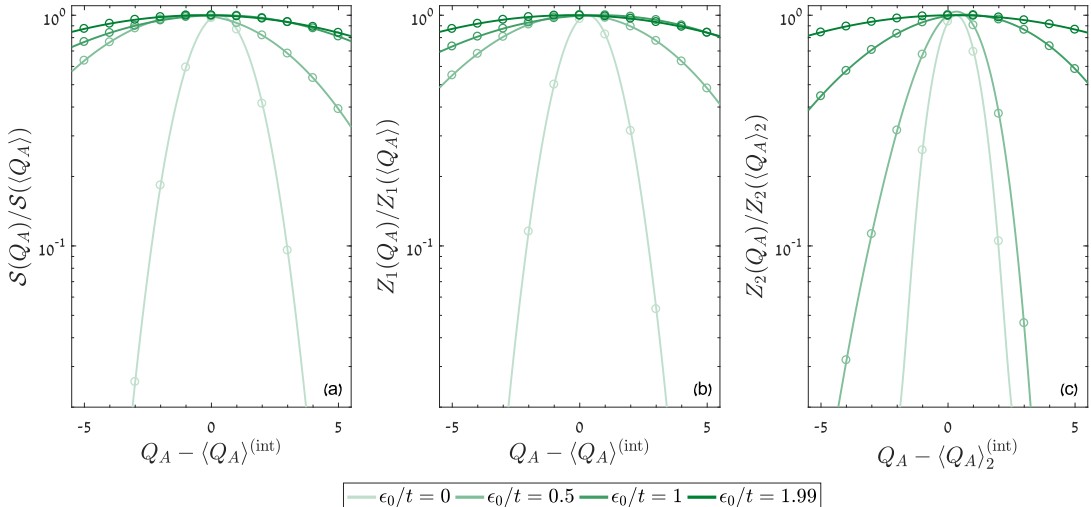

Figure 4: The single impurity model: Symmetry-resolved entanglement measures, calculated using Eqs. (19) and (20), where the generating function was evaluated analytically through Eq. (34) (lines) and numerically through Eqs. (17) and (21) (circles), in a subsystem of $L = 20000$ sites. Results are shown as a function of $Q_A$, the charge in subsystem $A$, for various fixed values of $\epsilon_0/t$, with $k_{F,R} = \pi/2$ and $k_{F,L} - k_{F,R} = 0.1$. The charge $Q_A$ is measured relative to the rounded mean charge $\langle Q_A \rangle_n^{(int)}$ defined in Eq. (54), and the results are normalized by the analytical value exactly at $\langle Q_A \rangle_n$. Panels (a) and (b) show the resolved vNEE and first Rényi moment, respectively, for which we set $n = 1$ (with $\langle Q_A \rangle = \langle Q_A \rangle_{n=1}$), whereas panel (c) shows the resolved second Rényi moment, where we use $n = 2$.

## 4.2 Charge-resolved entanglement

Next, we studied the symmetry resolution of the Rényi moments $Z_1$ and $Z_2$ and of the vNEE. This was done by extracting the symmetry-resolved quantities from both the analytical calculation (Eq. (34)) and the numerical calculation of the generating function $Z_n(\alpha)$, relying on Eqs. (19) and (20). The numerical estimation of $Z_n(\alpha)$ was obtained using Eq. (21), again using the approximation in Eq. (17) for the correlation matrix, which assumes an infinite distance between subsystem $A$ and the impurity.

The results are presented in Fig. 4, where it is evident that the numerical results (naturally sampled at integer values of $Q_A$) accurately fit the analytical results near the mean charges of the distributions, given by $\langle Q_A \rangle_n$ from Eq. (42) (with $n = 1$ for $\mathcal{S}(Q_A)$ and $Z_1(Q_A)$, further simplified in Eq. (39), and with $n = 2$ for $Z_2(Q_A)$). The plots in Fig. 4 are all centered around the integer charge that is the nearest to the corresponding mean charge, given by

$$\langle Q_A \rangle_n^{(int)} = \left\lceil \frac{1}{2} \lfloor 2 \langle Q_A \rangle_n \rfloor \right\rceil, \tag{54}$$

where $\lfloor \rfloor$ is the floor function ($\lfloor x \rfloor$ is the nearest integer to $x$ from below) and $\lceil \rceil$ is the ceiling function ($\lceil x \rceil$ is the nearest integer to $x$ from above). The charge-resolved quantities indeed reach their maximal value near the mean charge $\langle Q_A \rangle_n$, but due to their slight deviation from a Gaussian form (since $L$ is large but finite) the charge sector $Q_A = \langle Q_A \rangle_n^{(int)}$ is not necessarily the sector where they peak. Note that the mean charge $\langle Q_A \rangle_n$ varies with the model parameters $\epsilon_0/t$ and $k_\pm$.

A conspicuous property of the resolved quantities is that their distribution among charge sectors becomes wider as the model parameters approach values such that $|t_L(k)|^2 \approx |r_R(k)|^2$

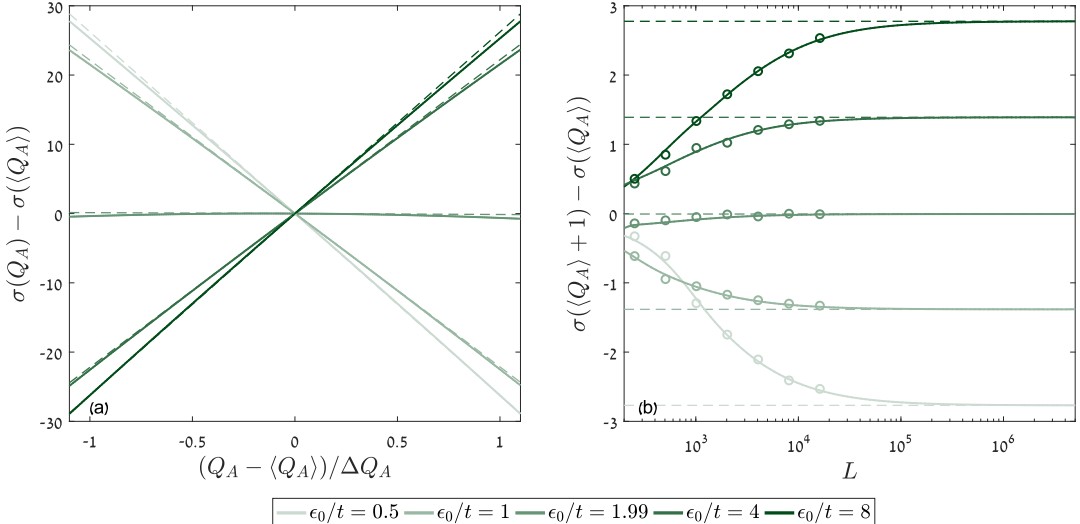

Figure 5: The single impurity model: Breaking of entanglement equipartition by the post-projection vNEE $\sigma(Q_A)$, calculated using Eqs. (5), (19) and (20). Results are shown for various fixed values of $\epsilon_0/t$, with $k_{F,R} = \pi/2$ and $k_{F,L} - k_{F,R} = 0.1$. In (a), $\sigma(Q_A)$ is calculated based on the analytically estimated generating function of Eq. (34), and measured relative to its estimated value at the mean charge $\langle Q_A \rangle$ for a subsystem of $L = 10^5$ sites (solid lines). Dashed lines designate the corresponding linear equipartition-breaking term that was approximated analytically for large $L$ in Eq. (50). In (b) we show the difference in $\sigma(Q_A)$ due to a shift of $Q_A$ in one charge unit near the mean charge $\langle Q_A \rangle$ as a function of the length $L$ of subsystem $A$. $\sigma(Q_A)$ is calculated both based on the analytically estimated generating function of Eq. (34) (solid lines), and based on the numerically estimated generating function of Eqs. (17) and (21) (circles; in the numerical case, $\sigma$ is estimated at the genuine charge sectors $\langle Q_A \rangle^{(\text{int})}$ and $\langle Q_A \rangle^{(\text{int})} + 1$ rather than at $\langle Q_A \rangle$ and $\langle Q_A \rangle + 1$). Dashed lines designate the $L$-independent slope of the linear equipartition-breaking term in Eq. (50).

for $k_- < k < k_+$. This is manifested in the analytical results most simply for $Z_1(Q_A)$, since the leading $\mathcal{O}(L)$ term in the analytical expression for the charge variance in Eq. (40) peaks exactly within that region in the space of the model parameters. In Fig. 4 we fixed $k_- = \pi/2$ and $\Delta k = 0.1$, so this condition is equivalent there to $\epsilon_0 \approx 2t$. The exact point $\epsilon_0 = 2t$ cannot be examined analytically due to a non-integrable divergence of the generating function, as explained in Subsec. 3.2. Nevertheless, the results for $\epsilon_0 = 1.99t$ in Fig. 4 indicate that, even at points very close to the specific point where the divergence occurs, this divergence does not cause any discernible deviation of the analytical calculation from numerical results.

Additionally, we examined the post-projection charge-resolved vNEE, $\sigma(Q_A)$, for the single impurity model. Since the analytical result of Eq. (49) relies on the Gaussian approximation of the generating function, it was natural to test whether it properly captured the behavior of the charge-resolved measures that were extracted from the more accurate analytical form of the generating function, given by Eq. (34). In Fig. 5 we present a comparison between the deviation from entanglement equipartition of the full analytical result and the leading-order term estimated in Eq. (50), which is linear in $Q_A - \langle Q_A \rangle$. From Fig. 5(a) it is evident that the linear breaking of entanglement equipartition holds up to $|Q_A - \langle Q_A \rangle| \approx \Delta Q_A$, confirming that the equipartition-breaking term scales as $\sqrt{L}$ for large $L$. Fig. 5(b) affirms that for large $L$, the slope of this linear term becomes independent of $L$.

Fig. 5(b) also includes fully numerical estimations (using the numerical calculation of

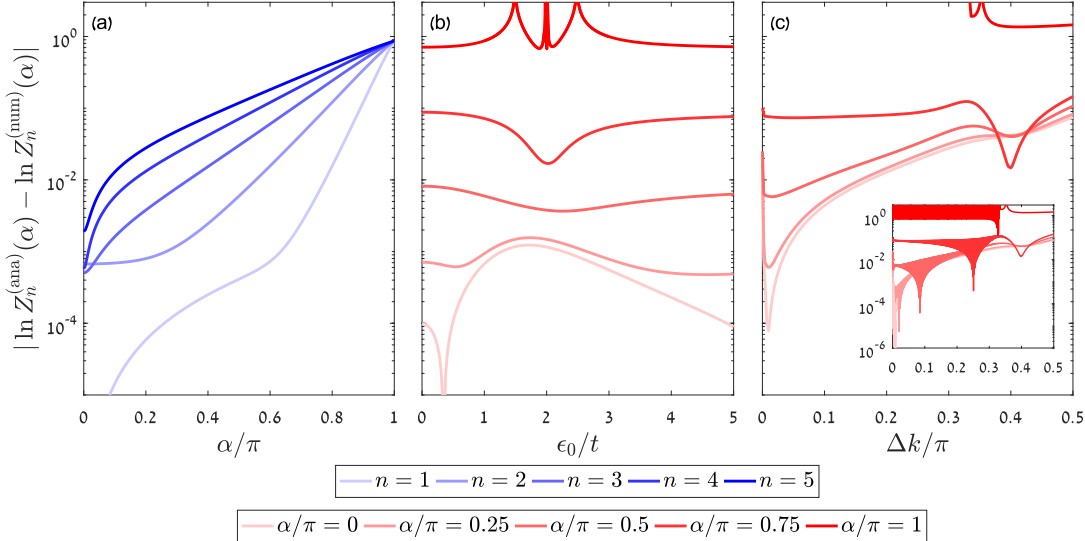

Figure 6: The single impurity model: The absolute deviation of the analytical result $\ln Z_n^{(ana)}(\alpha)$, given by Eq. (34), from the numerical result $\ln Z_n^{(num)}(\alpha)$ calculated using Eqs. (17) and (21), in a subsystem of $L = 1000$ sites. In the different panels, the absolute deviation is presented (a) as a function of $\alpha$ for various fixed values of $n$, with $\epsilon_0 = t$, $k_{F,R} = \pi/2$ and $k_{F,L} - k_{F,R} = 0.1$; (b) as a function of $\epsilon_0/t$ for various fixed values of $\alpha$, with $n = 2$, $k_{F,R} = \pi/2$ and $k_{F,L} - k_{F,R} = 0.1$; and (c) as a function of $\Delta k = k_{F,L} - k_{F,R}$ for various fixed values of $\alpha$, with $n = 2$, $\epsilon_0 = t$ and $k_{F,R} = \pi/2$. In (c) the results oscillate rapidly with varying $\Delta k$, so only the top envelope of these oscillations (designating the local maximum of the deviation) is shown, while the inset shows the full oscillations (for $\alpha = \pi$ there is a regime of $\Delta k$ for which at each period of the oscillation there is a point where $Z_n^{(num)}(\alpha = \pi)$ vanishes while $Z_n^{(ana)}(\alpha = \pi)$ does not, and therefore the top envelope is infinite and is not shown).

$Z_n(\alpha)$ and Eqs. (5), (19) and (20)) of the change in $\sigma(Q_A)$ between adjacent (integer valued) charge sectors near $\langle Q_A \rangle$, for reasonable subsystem sizes. These numerical results nicely follow the trend of their analytical counterparts, once again attesting to the validity of the latter.

## 4.3 Accuracy of the generating function calculation

With the generating function $Z_n(\alpha)$ being the basis for all the analytical calculations discussed in this paper, a test of the accuracy of its calculation across the parameter space is required. In Fig. 6 the analytical estimation of $\ln Z_n(\alpha)$ (denoted as $\ln Z_n^{(ana)}(\alpha)$) for the single impurity model is compared to a numerical calculation of $\ln Z_n(\alpha)$ (denoted as $\ln Z_n^{(num)}(\alpha)$). Here the numerical calculation once again neglects the effects of a finite distance between subsystem $A$ and the impurity (whose effects will be examined later on), relying on Eqs. (17) and (21). The comparison indicates good agreement between analytical and numerical results for values of $\alpha$ far enough from $\alpha = \pm\pi$. As explained above, the fact that this agreement breaks down as $\alpha$ approaches $\pm\pi$ is a well-known trait of the leading-order approximation that stems from the Fisher-Hartwig conjecture [67, 68, 70].

Importantly, Fig. 6 illustrates that for fixed values of $k_\pm$, divergences in the absolute deviation between analytical and numerical results may occur at $\alpha = \pm\pi$, though they are typically rare within the parameter space. Such singularities appear if either $\nu(k_-) = 0$, $\nu(k_+) = 0$ or $\nu_0 = 0$, where $\ln Z_n^{(ana)}(\alpha = \pm\pi)$ diverges (as discussed in Subsec. 3.2, see Eq. (35)), or

if one of the numerically calculated eigenvalues $\nu_l$ vanishes, thus setting the numerical result to $Z_n^{(\text{num})}(\alpha = \pm \pi) = 0$ according to Eq. (21). In Fig. 6(b), for example, singularities at $\alpha = \pi$ may be detected for values of $\epsilon_0$ near $2t$ (divergence of $\ln Z_n^{(\text{ana})}(\alpha)$) and near $1.5t, 2.5t$ (points where $Z_n^{(\text{num})}(\alpha)$ vanishes; the locations of these points in the parameter space varies with $L$). However, when varying $k_\pm$ and fixing all other parameters, $\ln Z_n^{(\text{num})}(\alpha)$ oscillates rapidly as a function of $\Delta k$, with periodicity $2\pi/L$. These oscillations are not reflected in the analytical result, causing its deviation from $\ln Z_n^{(\text{num})}(\alpha)$ to oscillate rapidly as well, as seen in Fig. 6(c). Fig. 6(c) more specifically indicates that for $\alpha = \pm \pi$, there exists a regime of values of $k_\pm$ where in each period of the oscillation there is a singularity of the deviation. Each such singularity corresponds to a vanishing numerically calculated eigenvalue $\nu_l$, i.e. to a zero of $Z_n^{(\text{num})}(\alpha = \pm \pi)$.

Finally, we address the effect of a finite distance between subsystem $A$ and the impurity – i.e., of the Hankel term in Eq. (13) that was omitted from Eq. (17), and was heretofore disregarded. Although we were not able to incorporate the effect of the Hankel term into our analytical calculation, the concrete example of the single impurity model allows us to examine numerically the dependence of the results on $d$, the distance of $A$ from the impurity at the origin. More precisely, $d$ is taken to be the location of the leftmost site in subsystem $A$, such that $A$ includes the sites $n = d, \ldots, d + L - 1$. Fig. 7 shows the comparison between two numerical calculations of the generating function: the one extracted from the approximate form of the correlation matrix in Eq. (17) (denoted as $Z_n^{(\infty)}(\alpha)$), which was used in the comparison to the analytical results, and the one that relies on the full form of the correlation matrix in Eq. (13) (denoted as $Z_n^{(d)}(\alpha)$).

One may observe that the additional term that accounts for finite $d$ effects on $\ln Z_n(\alpha)$ oscillates as a function of $d$, a manifestation of Friedel oscillations [101, 102]. The typical wavenumber of these oscillations is $2k_{F,R}$, and is therefore independent of $k_{F,L}$, as should be expected from the fact that the $d$-dependent term in Eq. (13) is independent of $k_{F,L}$ as well (this holds for a subsystem to the right of the scattering region; for a subsystem on the left, the roles of $k_{F,L}$ and $k_{F,R}$ are switched). The numerical results also verify that the deviation of $\ln Z_n^{(d)}(\alpha)$ from $\ln Z_n^{(\infty)}(\alpha)$ indeed vanishes as $d \to \infty$. Furthermore, when averaging over the oscillations, one finds that for $d \gg L$ their average approaches a power law decay proportional to $d^{-2}$, while their amplitude exhibits a power law behavior proportional to $d^{-1}$, typical of Friedel oscillations in 1D[3] [101]. This observation dovetails with the aforementioned projection of an algebraic decay of the Hankel term in the correlation matrix.

The dependence of the generating function on $d$ may be seen as an effect of boundary conditions, referring here to the boundary of subsystem $A$. Finite $d$ effects are therefore expected not to be reflected in the leading, linear in $L$ term of $\ln Z_n(\alpha)$, since this term represents an extensive property of $A$. We have verified numerically that the Hankel contribution indeed has no extensive effect. The logarithmic term of $\ln Z_n(\alpha)$, in contrast, does generally depend on the boundary conditions of $A$ (cf. Refs. [51, 109]), and so for finite $d$ the Hankel contribution may be proportional to $\ln L$, though such a logarithmic dependence is in practice hard to ascertain numerically for accessible subsystem sizes. Regardless of this, considering that in the previous calculations we kept terms which are constant in $L$, and that the Hankel contribution is proportional to $d^{-1}$ provided that $d \gg L$, neglecting the $d$ dependence is certainly justified in the regime $d \gg L$.

---

[3]In $D$ dimensions, Friedel oscillations tend to decay as $1/R^D$, where $R$ is the distance from the impurity.

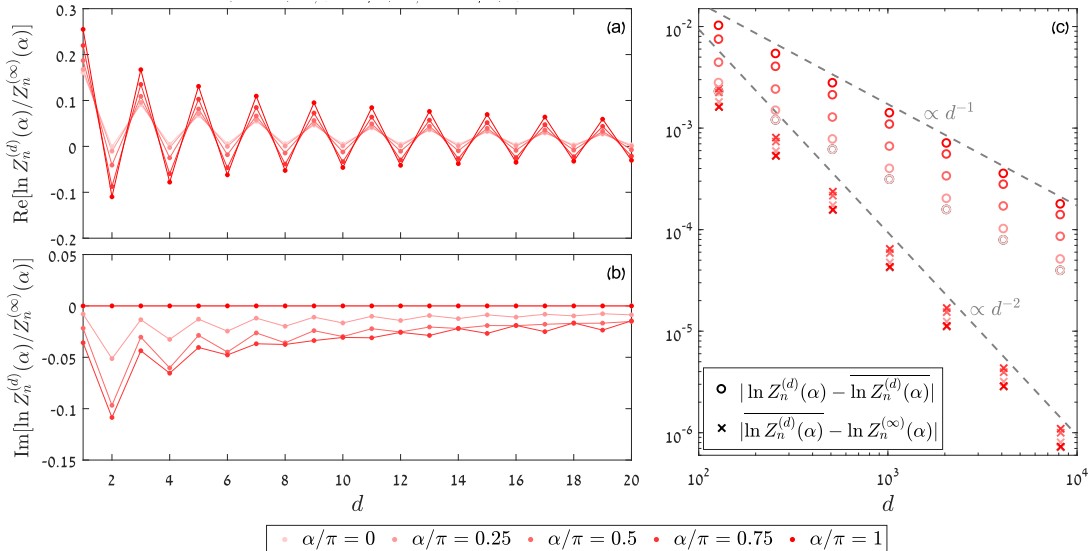

Figure 7: The single impurity model: The deviation of $\ln Z_n^{(d)}(\alpha)$, the distance-dependent generating function calculated numerically using Eqs. (13) and (21) (where $d$ is the distance between the leftmost site of $A$ and the origin, at which the scatterer is located), from $\ln Z_n^{(\infty)}(\alpha)$, the generating function for $d \to \infty$ calculated numerically using Eqs. (17) and (21), in a subsystem of $L = 100$ sites. Results are shown for various fixed values of $\alpha$, with $n = 2$, $\epsilon_0 = t$, $k_{F,R} = \pi/2$ and $k_{F,L} = 2\pi/3$. Both (a) the real part and (b) the imaginary part of the deviation are plotted (dots), accompanied by thin lines as guides to the eye (note that for $\alpha = 0$ the imaginary part vanishes by definition). Panel (c) shows the deviation for $d > L$ following averaging over oscillations, where the absolute values of both the average $\overline{\ln Z_n^{(d)}(\alpha)} - \ln Z_n^{(\infty)}(\alpha)$ and the amplitude $\ln Z_n^{(d)}(\alpha) - \overline{\ln Z_n^{(d)}(\alpha)}$ of the oscillations are plotted. Dashed gray lines emphasize that for all plotted values of $\alpha$, the average deviation approaches a $\propto d^{-2}$ power law behavior and the amplitude approaches a $\propto d^{-1}$ power law behavior.

## 5 Generalization to multiple scatterers

In the following section we rely on the analytical results for the model described in Sec. 2 in order to derive corresponding results for a more general scenario, where the tight-binding chain contains several different scattering regions rather than just a single one. The necessary foundation is the description of the combined scattering effects of two scatterers, with a distance of $\ell$ sites between them. We assume throughout this section that $\ell$ is considerably larger than the Fermi wavelengths, $(k_{F,L})^{-1}$ and $(k_{F,R})^{-1}$. We mark the left scatterer with I, and the right one with II.

Let us associate a unitary scattering matrix with each scattering region,

$$S_i(k) = \begin{pmatrix} r_L^{(i)}(k) & t_R^{(i)}(k) \\ t_L^{(i)}(k) & r_R^{(i)}(k) \end{pmatrix}, \tag{55}$$

where $i = \text{I}, \text{II}$. If subsystem $A$ is situated to the same side of both scattering regions, as depicted in Fig. 8(a), we may treat them as a single scatterer with appropriate reflection and

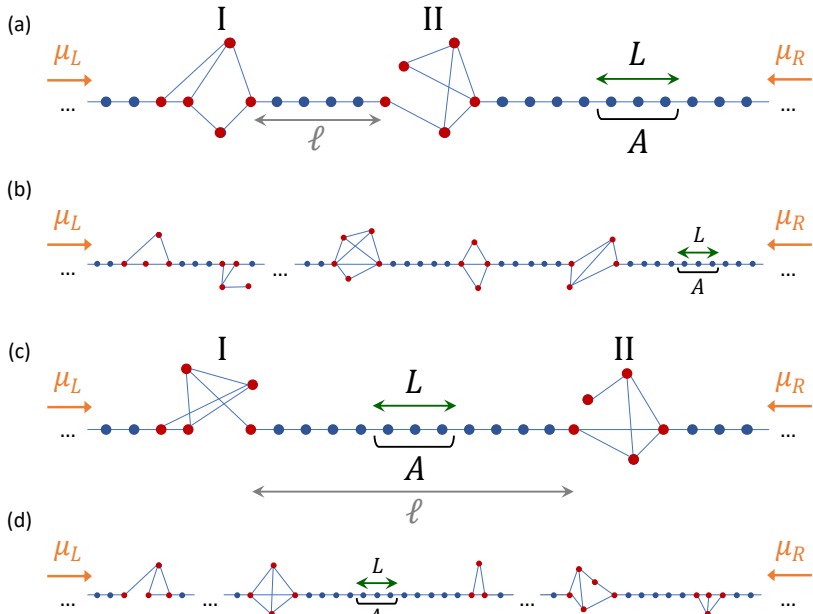

Figure 8: Schematic illustrations of the lattice models containing multiple scattering regions, which are discussed throughout Sec. 5. Sites marked in blue belong to the unperturbed parts of the tight-binding chain, while sites marked in red belong to the scattering regions. $\mu_L$ ($\mu_R$) designates the chemical potential for particles incoming from the left (right). $A$ denotes the subsystem of $L$ contiguous sites with respect to which the calculations of bipartite entanglement are performed. Panel (a) depicts a scenario where two scattering regions are located on the same side of $A$. Panel (b) depicts a more general scenario with multiple scattering regions, that are all on the same side of $A$. Panel (c) depicts a scenario where $A$ is located between two scattering regions. Panel (d) depicts an example for the general scenario, with multiple scattering regions on both sides of $A$. In (a) and (c), $\ell$ denotes the number of sites between scattering regions I and II, which is assumed to be much larger than the Fermi wavelengths.

transmission amplitudes [89]. For a wave incoming from the left, these are given by

$$r_L^{(\mathrm{I+II})} = r_L^{(\mathrm{I})} + \frac{t_R^{(\mathrm{I})} t_L^{(\mathrm{I})} r_L^{(\mathrm{II})}}{e^{-2ik\ell} - r_R^{(\mathrm{I})} r_L^{(\mathrm{II})}}, \quad t_L^{(\mathrm{I+II})} = \frac{t_L^{(\mathrm{I})} t_L^{(\mathrm{II})}}{1 - e^{2ik\ell} r_R^{(\mathrm{I})} r_L^{(\mathrm{II})}}, \tag{56}$$

and for a wave incoming from the right the amplitudes $r_R^{(\mathrm{I+II})}$ and $t_R^{(\mathrm{I+II})}$ are given by the same expressions, up to exchanging $R \leftrightarrow L, \mathrm{I} \leftrightarrow \mathrm{II}$ and multiplying by an overall phase (the notation emphasizing the dependence of the scattering amplitudes on $k$ has been omitted for brevity).

As described in Sec. 3, the correlation matrix $C_A$ is the basis for our calculation of entanglement measures. By the same line of argument of Sec. 2, it can be written as a sum of a Toeplitz matrix and a Hankel matrix, where the Hankel matrix can be neglected assuming $A$ is far enough from the combined scatterer, as before. For the estimation of the Toeplitz term, we define the following incoherent scattering probabilities [89]:

$$|\bar{r}_L|^2 = \left|r_L^{(\mathrm{I})}\right|^2 + \frac{\left|t_R^{(\mathrm{I})} t_L^{(\mathrm{I})} r_L^{(\mathrm{II})}\right|^2}{1 - \left|r_R^{(\mathrm{I})} r_L^{(\mathrm{II})}\right|^2}, \quad |\bar{t}_L|^2 = \frac{\left|t_L^{(\mathrm{I})} t_L^{(\mathrm{II})}\right|^2}{1 - \left|r_R^{(\mathrm{I})} r_L^{(\mathrm{II})}\right|^2}. \tag{57}$$

The probabilities $|\overline{r}_L(k)|^2$ and $|\overline{t}_L(k)|^2$ differ from the expressions $\left|r_L^{(\text{I+II})}(k)\right|^2$ and $\left|t_L^{(\text{I+II})}(k)\right|^2$ (respectively) by terms that oscillate as $e^{2ik\ell}$, and that (recalling the assumption $\ell \gg (k_{F,L})^{-1}, (k_{F,R})^{-1}$) can thus be neglected when integrating over $k$. Analogous definitions of probabilities $|\overline{r}_R(k)|^2$ and $|\overline{t}_R(k)|^2$ (again applying the replacements $R \leftrightarrow L, \text{I} \leftrightarrow \text{II}$ in Eq. (57)) are used to replace $\left|r_R^{(\text{I+II})}(k)\right|^2$ and $\left|t_R^{(\text{I+II})}(k)\right|^2$, respectively, under integrals over $k$. This finally allows us to write the two-point correlation matrix of $A$ according to Eq. (17), where the incoherent scattering probabilities stand in for the original scattering probabilities of a single scattering region. This scheme can be readily extended to treat multiple scattering regions that are all located on the same side of subsystem $A$ (as illustrated in Fig. 8(b)) as a single combined scattering region, and therefore one may use the analytical results of Sec. 3 to estimate the various entanglement measures discussed there.

Considering the case where scattering regions are located on both sides of subsystem $A$ (depicted generally in Fig. 8(d)), this scheme of calculating combined scattering probabilities has reduced the problem to that of two scattering regions – region I to the left of $A$, and region II to its right, as illustrated in Fig. 8(c). The distances from the edges of $A$ to the scatterers are assumed to be much larger than $L$, the number of sites in $A$. Using the assumption of the distance between scatterers I and II being much larger than the Fermi wavelengths, we arrive at the following approximation of the correlation matrix in that region:

$$C_{mn} \approx \frac{1}{2\pi} \int_{-\pi}^{\pi} e^{-i(m-n)k} \overline{\tau}(k)\,dk\,. \tag{58}$$

Here we have defined

$$\overline{\tau}(k) = \begin{cases} 1 & -k_- < k < k_-\,, \\ \frac{1}{2}\left(1 + \overline{\nu}_{\text{I}}(k)\right) & k_- < k < k_+\,, \\ \frac{1}{2}\left(1 - \overline{\nu}_{\text{II}}(-k)\right) & -k_+ < k < -k_-\,, \\ 0 & \text{otherwise}\,, \end{cases} \tag{59}$$

where

$$\frac{1 \pm \overline{\nu}_{\text{I}}(k)}{2} = \frac{\left|t_L^{(\text{I})}(k)\right|^2}{1 - \left|r_R^{(\text{I})}(k)\,r_L^{(\text{II})}(k)\right|^2} \quad \text{for } k_{F,L} = k_\pm\,, \tag{60}$$

and $\overline{\nu}_{\text{II}}(k)$ is defined similarly, up to replacing $\left|t_L^{(\text{I})}(k)\right|^2$ with $\left|t_R^{(\text{II})}(k)\right|^2$. The derivation of Eq. (58) is detailed in Appendix A.5.

The analytical method of Subsec. 3.1 can now be applied to obtain the exact form of the two leading terms in the asymptotic expression for the generating function defined in Eq. (18):

$$\ln Z_n(\alpha) \sim \frac{L}{2\pi}\left[2i\alpha k_- + \int_{k_-}^{k_+} \left\{e_n^{(\alpha)}(1, \overline{\nu}_{\text{I}}(k)) + e_n^{(\alpha)}(1, -\overline{\nu}_{\text{II}}(k))\right\}dk\right]$$
$$+ \ln L\left[Q_n(\overline{\nu}_{\text{I}}(k_-), \alpha) + Q_n(-\overline{\nu}_{\text{I}}(k_+), -\alpha)\right]$$
$$+ \ln L\left[Q_n(-\overline{\nu}_{\text{II}}(k_-), \alpha) + Q_n(\overline{\nu}_{\text{II}}(k_+), -\alpha)\right]\,. \tag{61}$$

An approximation for the first subleading correction to this asymptotics, which is independent of $L$, may be derived by following the analytical method of Subsec. 3.2. The various entanglement measures that were discussed in the context of a single scattering region can now be extracted from Eq. (61).

We report in particular the result for the unresolved vNEE in subsystem $A$ between the two scatterers. For convenience, we introduce the notations

$$\mathcal{T}_{\mathrm{I}}(k) = \frac{\left|t_L^{(\mathrm{I})}(k)\right|^2}{1 - \left|r_R^{(\mathrm{I})}(k)\, r_L^{(\mathrm{II})}(k)\right|^2} \quad , \quad \mathcal{T}_{\mathrm{II}}(k) = \frac{\left|t_R^{(\mathrm{II})}(k)\right|^2}{1 - \left|r_R^{(\mathrm{I})}(k)\, r_L^{(\mathrm{II})}(k)\right|^2}. \tag{62}$$

The vNEE is given by the asymptotic form

$$
\begin{aligned}
\mathcal{S} \sim & -\frac{L}{2\pi}\int_{k_-}^{k_+} \left[\mathcal{T}_{\mathrm{I}}(k)\ln \mathcal{T}_{\mathrm{I}}(k) + (1-\mathcal{T}_{\mathrm{I}}(k))\ln(1-\mathcal{T}_{\mathrm{I}}(k))\right] dk \\
& -\frac{L}{2\pi}\int_{k_-}^{k_+} \left[\mathcal{T}_{\mathrm{II}}(k)\ln \mathcal{T}_{\mathrm{II}}(k) + (1-\mathcal{T}_{\mathrm{II}}(k))\ln(1-\mathcal{T}_{\mathrm{II}}(k))\right] dk \\
& + \left[q\left(\mathcal{T}_{\mathrm{I}}\left(k_{F,R}\right)\right) + q\left(1-\mathcal{T}_{\mathrm{I}}\left(k_{F,L}\right)\right) + q\left(\mathcal{T}_{\mathrm{II}}\left(k_{F,L}\right)\right) + q\left(1-\mathcal{T}_{\mathrm{II}}\left(k_{F,R}\right)\right)\right]\ln L + \mathcal{O}(1),
\end{aligned}
\tag{63}
$$

where the function $q(p)$ was defined in Eq. (43). Note that if we require transmission and reflection factors to be constant functions of $k$, we recreate Eq. (26) of Ref. [31]. We have therefore generalized the scenario discussed in Ref. [31], where the subsystem lies on a tight binding chain coupled to two macroscopic leads with $k$-independent hybridization factors.

# 6 Conclusions and outlook

While the exact physical description of nonequilibrium many-body states remains a coveted yet elusive goal, entanglement measures continue to facilitate incremental progress toward its achievement. In this work we sought to exactly quantify the steady-state entanglement of a paradigmatic 1D lattice model, namely a homogeneous tight-binding chain interrupted by an arbitrary number of scattering regions, held under a bias voltage at zero temperature. For this purpose we employed the generalized Fisher-Hartwig conjecture to calculate bipartite entanglement measures for a subsystem located far away from the scatterers.

A central result of our work is given in Eq. (44). The von-Neumann entanglement entropy was shown to scale extensively with the size of the subsystem in question, with an additive logarithmic correction that arises from the sharp jumps in the energy distribution. While in the ground state such a scaling law is considered exotic and requires long range couplings [52,91], the class of steady states we investigated, which are excited eigenstates of the Hamiltonian with a Fermi-discontinuous distribution of the excitations [52], exhibits it generically. This suggests that a scaling law of the form of Eq. (44) should be much more common in nonequilibrium setups, rendering it a strong signature of the unique properties that distinguish steady states in and out of equilibrium. More precisely, this entanglement scaling law should in general be observed in steady states where the single-particle energy distribution features partially occupied states (of non-vanishing measure) and Fermi discontinuities. Refs. [30, 31] have indeed found such scaling of the vNEE for nonequilibrium steady states in some particular impurity setups. We expect this behavior to apply also to other current-carrying impurity models at zero temperature, including systems that contain massless Dirac fermions[4] or host dissipative defects [110, 111].

---

[4]Indeed, in the regime of a small bias voltage in our model, the dispersion relation for states within the voltage window may be linearized, giving an effective description of them as Dirac fermion states. These states are those responsible for the emergence of the linear and logarithmic leading terms in Eq. (44).

Notably, the form of the extensive term of the vNEE encapsulates the basic elements defining the steady state. It arises from scattering states within the energy window between the two different chemical potentials, as scattered particles traverse the subsystem from side to side, and thereby entangle its entire bulk to the rest of the chain. The assumption of a large subsystem allows us to disregard boundary effects and attribute classical probabilities to the scattering processes, and therefore the contribution of each mode to the entanglement is equivalent to a classical mixture entropy. As a consequence, the extensive term of the vNEE vanishes either in the absence of a bias voltage (i.e., in equilibrium), or if the scattering region is trivial. In this sense, the model studied here can be seen as a minimal model for producing such scaling of the steady state entanglement. The picture of entanglement as a result of partial occupation of momentum states due to scattering is also reflected in the linear term of $\ln Z_n(\alpha)$ in Eq. (27), which generalizes the known full counting statistics formula found by Levitov and Lesovik [99, 112, 113].

The exact expression for the vNEE is only one out of the comprehensive set of results presented in this paper, all encoded in the asymptotics of the generating function $Z_n(\alpha)$ in Eq. (34). These results include Rényi moments (from which Rényi entropies are readily obtained), statistical charge properties, and charge-resolved moments and entanglement measures. We particularly emphasize the novelty of the result in Eq. (49) for the vNEE following a projective charge measurement $\sigma(Q_A)$. It implies that to leading order, entanglement is equally distributed across charge sectors, as has been established for the great majority of models for which symmetry-resolved entanglement was studied. However, the term breaking entanglement equipartition grows with the subsystem size $L$, as $\sigma(Q_A) - \mathcal{S} = \mathcal{O}(\sqrt{L})$, for charge sectors within the standard deviation from the mean charge. The leading equipartition-breaking term was also found to be anti-symmetric in $Q_A - \langle Q_A \rangle$.

To the best of our knowledge, the model studied in this paper is the first where $\sigma(Q_A)$ exhibits these properties. A natural question for future research is therefore whether this unique behavior can be exclusively ascribed to the partially-transmitted current carried by the nonequilibrium steady state, as it was not witnessed in a different nonequilibrium model where the net current had been absent [73]. We additionally note that we expect the new behaviors uncovered in this work to hold in the presence of interactions, although further investigation is required to establish this claim.

Our analytical results were tested against numerics in a model where a single impurity site serves as the scatterer, and were shown to compare to them favorably. We additionally used numerics to observe effects of a finite distance between the subsystem and the scattering region, detecting signatures of Friedel oscillations and confirming that the effects are indeed negligible when that distance is large enough. We note that by using various proposed protocols for the measurement of (resolved and unresolved) entanglement measures [61–64, 84], our results may be experimentally tested in setups based on cold atom or electronic systems [89, 90].

The road toward a deeper understanding of nonequilibrium many-body physics is still riddled with unanswered questions. Hopefully the exact results presented in this paper can serve as building blocks for the future description of richer and more intricate phenomena out of equilibrium, such as effects of interactions, disorder, localization or external driving [53, 114, 115], entanglement phase transitions [92, 116–118], and transport through mesoscopic systems [119, 120].

## Acknowledgments

We thank P. Calabrese, M. Dalmonte and E. Sela for stimulating discussions. Our work has been supported by the U.S.-Israel Binational Science Foundation (Grant No. 2016224).

# Appendix

## A  Detailed derivations

In what follows we expand on the derivations of central analytical results discussed in Secs. 3 and 5. These analytical results include the expressions for the logarithmic and constant terms in the asymptotics of $\ln Z_n(\alpha)$ (Appendices A.1 and A.2); the expansion of $\ln Z_1(\alpha)$ in powers of $\alpha$ (Appendix A.3), from which statistical charge properties were extracted in Subsec. 3.3; the unresolved vNEE in the presence of a single scatterer (Appendix A.4); and the generalized form of the correlation matrix for the case of a subsystem between two scatterers (Appendix A.5).

### A.1  Logarithmic term in $\ln Z_n(\alpha)$

We present here a detailed derivation of the term $\mathcal{I}_{\log}(n, \alpha)$ in the asymptotic form (27) of $\ln Z_n(\alpha)$. Let us start by defining the complex function $\beta_{a,b}(\lambda) = \frac{1}{2\pi i} \ln \frac{\lambda-b}{\lambda-a}$ for two real numbers $a < b$, choosing the principal branch of the logarithm, such that $\left|\mathrm{Re}\,\beta_{a,b}(\lambda)\right| < \frac{1}{2}$. A crucial property of this function is that for a real $x \neq a, b$,

$$\beta_{a,b}\left(x + i0^{\pm}\right) = \begin{cases} \frac{1}{2\pi i} \ln \left|\frac{x-b}{x-a}\right| \pm \frac{1}{2} & x \in (a, b) \,, \\ \frac{1}{2\pi i} \ln \left|\frac{x-b}{x-a}\right| & x \notin [a, b] \,. \end{cases} \tag{A.1}$$

Using Eq. (22) and the asymptotic expression for $\ln D_L(\lambda)$ in Eq. (26), we may write

$$\mathcal{I}_{\log}(n, \alpha) = \lim_{\varepsilon, \delta \to 0^+} \frac{1}{2\pi i} \int\limits_{c(\varepsilon, \delta)} \ln L \left( \beta_{\nu(k_-), 1}(\lambda)^2 + \beta_{-1, \nu(k_+)}(\lambda)^2 + \beta_{-1, 1}(\lambda)^2 \right)$$

$$\times \frac{d}{d\lambda} e_n^{(\alpha)}(1 + \varepsilon, \lambda)\, d\lambda, \tag{A.2}$$

where we have employed integration by parts. Using the property from Eq. (A.1) and taking the limit $\varepsilon, \delta \to 0^+$, we obtain

$$\mathcal{I}_{\log}(n, \alpha) = \frac{1}{2\pi^2} \left[ \int\limits_{\nu(k_-)}^{1} \ln \left|\frac{x-1}{x-\nu(k_-)}\right| + \int\limits_{-1}^{\nu(k_+)} \ln \left|\frac{x-\nu(k_+)}{x+1}\right| + \int\limits_{-1}^{1} \ln \left|\frac{x-1}{x+1}\right| \right] \frac{d}{dx} e_n^{(\alpha)}(1, x)\, dx. \tag{A.3}$$

Through a change of variables $x \to -x$ we may notice that

$$\int\limits_{-1}^{\nu} \ln \left|\frac{x-\nu}{x+1}\right| \frac{d}{dx} e_n^{(\alpha)}(1, x)\, dx = \int\limits_{-\nu}^{1} \ln \left|\frac{x-1}{x+\nu}\right| \frac{d}{dx} e_n^{(-\alpha)}(1, x)\, dx \,, \tag{A.4}$$

and therefore

$$\mathcal{I}_{\log}(n, \alpha) = Q_n(\nu(k_-), \alpha) + Q_n(-\nu(k_+), -\alpha) + \frac{1}{12} \left( \frac{1}{n} - n \right) - \frac{\alpha^2}{4\pi^2 n}. \tag{A.5}$$

Here we invoked the notation from Eq. (28), and the term written explicitly was obtained by carrying out the integration through the change of variables $u = \ln \left|\frac{x-1}{x+1}\right|$.

## A.2  Subleading term in $\ln Z_n(\alpha)$

We detail the calculation of the expression in Eq. (32) for the term $\tilde{\mathcal{I}}_{\text{const}}(n,\alpha)$, which is the approximate subleading term in the asymptotics of $\ln Z_n(\alpha)$ in Eq. (34). The approximate piecewise-constant symbol $\tilde{\phi}(k)$ of Eq. (29) can be written in a Fisher-Hartwig form [88]:

$$\tilde{\phi}(k) = E(\lambda) \cdot e^{i\sum_{j=1}^3 (k-k_j)\beta_j} \prod_{j=1}^3 g_j(k), \tag{A.6}$$

where $k_1 \equiv k_-, \beta_1 \equiv \beta_{\nu_0,1}(\lambda), k_2 \equiv k_+, \beta_2 \equiv \beta_{-1,\nu_0}(\lambda), k_3 \equiv 2\pi - k_{F,R}, \beta_3 \equiv -\beta_{-1,1}(\lambda)$ (the function $\beta_{a,b}(\lambda)$ is defined in Appendix A.1), and where we also defined

$$E(\lambda) = (\lambda-1)^{(k_-+k_{F,R})/2\pi}(\lambda-\nu_0)^{\Delta k/2\pi}(\lambda+1)^{1-(k_{F,R}+k_+)/2\pi}, \tag{A.7}$$

and

$$g_j(k) = \begin{cases} e^{i\pi\beta_j} & 0 \le k < k_j, \\ e^{-i\pi\beta_j} & k_j \le k < 2\pi. \end{cases} \tag{A.8}$$

According to the Fisher-Hartwig conjecture, the asymptotics of the Toeplitz determinant $\tilde{D}_L(\lambda)$, arising from this symbol, is given by [88, 103]

$$\tilde{D}_L(\lambda) \sim E(\lambda)^L L^{-\sum_{j=1}^3 \beta_j^2} \times \prod_{1 \le j < m \le 3} \left| e^{ik_j} - e^{ik_m} \right|^{2\beta_j\beta_m} \prod_{j=1}^3 G(1+\beta_j) G(1-\beta_j), \tag{A.9}$$

where $G(x)$ is the Barnes G-function [121], which obeys in particular

$$G(1+z) = \Gamma(z) G(z). \tag{A.10}$$

The required subleading term $\tilde{\mathcal{I}}_{\text{const}}(n,\alpha)$ in Eq. (30) is therefore given by the integral

$$\tilde{\mathcal{I}}_{\text{const}}(n,\alpha) = -\lim_{\varepsilon,\delta\to 0^+} \frac{1}{2\pi i} \int_{c(\varepsilon,\delta)} \ln\left[\frac{\tilde{D}_L(\lambda)}{E(\lambda)^L L^{-\sum_{j=1}^3 \beta_j^2}}\right] \frac{d}{d\lambda} e_n^{(\alpha)}(1+\varepsilon,\lambda) d\lambda. \tag{A.11}$$

Let us now denote

$$\omega_{jm} = -\lim_{\varepsilon,\delta\to 0^+} \frac{1}{2\pi i} \int_{c(\varepsilon,\delta)} \ln\left[\left| e^{ik_j} - e^{ik_m}\right|^{2\beta_j\beta_m}\right] \frac{d}{d\lambda} e_n^{(\alpha)}(1+\varepsilon,\lambda) d\lambda. \tag{A.12}$$

Then, relying on the property from Eq. (A.1), we obtain

$$\omega_{12} = -\frac{\ln\left|2\sin\left(\frac{1}{2}\Delta k\right)\right|}{2\pi^2}\left[\int_{-1}^{\nu_0} \ln\left|\frac{x-1}{x-\nu_0}\right| \frac{d}{dx} e_n^{(\alpha)}(1,x)\,dx + \int_{\nu_0}^1 \ln\left|\frac{x-\nu_0}{x+1}\right| \frac{d}{dx} e_n^{(\alpha)}(1,x)\,dx\right],$$

$$\omega_{13} = \frac{\ln\left|2\sin\left(\frac{k_-+k_{F,R}}{2}\right)\right|}{2\pi^2}\left[\int_{-1}^1 \ln\left|\frac{x-1}{x-\nu_0}\right| \frac{d}{dx} e_n^{(\alpha)}(1,x)\,dx + \int_{\nu_0}^1 \ln\left|\frac{x-1}{x+1}\right| \frac{d}{dx} e_n^{(\alpha)}(1,x)\,dx\right],$$

$$\omega_{23} = \frac{\ln\left|2\sin\left(\frac{k_++k_{F,R}}{2}\right)\right|}{2\pi^2}\left[\int_{-1}^{\nu_0} \ln\left|\frac{x-1}{x+1}\right| \frac{d}{dx} e_n^{(\alpha)}(1,x)\,dx + \int_{-1}^1 \ln\left|\frac{x-\nu_0}{x+1}\right| \frac{d}{dx} e_n^{(\alpha)}(1,x)\,dx\right],$$

$$\tag{A.13}$$

and thus, when we sum up the different contributions, we have

$$
\begin{aligned}
\omega_{12} + \omega_{13} + \omega_{23} = {} & \ln\left| \frac{2\sin\left(\frac{k_- + k_{F,R}}{2}\right)\sin\left(\frac{1}{2}\Delta k\right)}{\sin\left(\frac{k_+ + k_{F,R}}{2}\right)} \right| Q_n\left(\nu_0, \alpha\right) \\
& + \ln\left| \frac{2\sin\left(\frac{k_+ + k_{F,R}}{2}\right)\sin\left(\frac{1}{2}\Delta k\right)}{\sin\left(\frac{k_- + k_{F,R}}{2}\right)} \right| Q_n\left(-\nu_0, -\alpha\right) \\
& + \ln\left| \frac{2\sin\left(k_{F,R}\right)\sin\left(k_0\right)}{\sin\left(\frac{1}{2}\Delta k\right)} \right| \left[\frac{1}{12}\left(\frac{1}{n} - n\right) - \frac{\alpha^2}{4\pi^2 n}\right],
\end{aligned}
\tag{A.14}
$$

employing the notation from Eq. (28).

Let us further denote

$$
\rho_j = -\lim_{\varepsilon,\delta\to 0^+} \frac{1}{2\pi i} \int_{c(\varepsilon,\delta)} \ln\left[G\left(1+\beta_j\right)G\left(1-\beta_j\right)\right] \frac{d}{d\lambda} e_n^{(\alpha)}(1+\varepsilon,\lambda)\, d\lambda,
\tag{A.15}
$$

and examine, for example, $\rho_1$. Then, using Eq. (A.1), we have

$$
\rho_1 = \frac{1}{2\pi i} \int_{\nu_0}^{1} \ln\left[ \frac{G\left(\frac{3}{2} + \frac{1}{2\pi i}\ln\left|\frac{x-1}{x-\nu_0}\right|\right) G\left(\frac{1}{2} - \frac{1}{2\pi i}\ln\left|\frac{x-1}{x-\nu_0}\right|\right)}{G\left(\frac{1}{2} + \frac{1}{2\pi i}\ln\left|\frac{x-1}{x-\nu_0}\right|\right) G\left(\frac{3}{2} - \frac{1}{2\pi i}\ln\left|\frac{x-1}{x-\nu_0}\right|\right)} \right] \frac{d}{dx} e_n^{(\alpha)}(1,x)\, dx,
\tag{A.16}
$$

and by applying Eq. (A.10) we obtain $\rho_1 = \Upsilon_n\left(\nu_0, \alpha\right)$, per the notation in Eq. (33). In a similar manner we find that $\rho_2 = \Upsilon_n\left(-\nu_0, -\alpha\right)$ and $\rho_3 = \Upsilon_n\left(-1, \alpha\right)$. Finally, since

$$
\tilde{\mathcal{I}}_{\mathrm{const}}(n,\alpha) = \sum_{1\le j < m \le 3} \omega_{jm} + \sum_{j=1}^{3} \rho_j,
\tag{A.17}
$$

we arrive at the desired result in Eq. (32).

## A.3 Expansion of the generating function for $n = 1$

Here we explicitly calculate the terms of order $\alpha$ and $\alpha^2$ in the power series expansion of $\ln Z_1(\alpha)$. We first note that

$$
e_1^{(\alpha)}(1,x) = \frac{i}{2}(1+x)\alpha + \frac{x^2-1}{8}\alpha^2 + \mathcal{O}\left(\alpha^3\right),
\tag{A.18}
$$

and therefore the linear term in Eq. (27) obeys

$$
\mathcal{I}_{\mathrm{lin}}(1,\alpha) = \frac{1}{2\pi}\left[ i\alpha\left(k_0 + k_{F,R}\right) + \int_{k_-}^{k_+} \left(\frac{i\alpha}{2}\nu(k) + \frac{\nu(k)^2 - 1}{8}\alpha^2\right) dk \right] + \mathcal{O}\left(\alpha^3\right).
\tag{A.19}
$$

For the logarithmic and constant terms we use the fact that $Q_1(\nu,\alpha)$ may be calculated explicitly for any $-1 \le \nu \le 1$. Indeed, the change of variables $u = \frac{x-\nu}{1-x}$ allows us to write

$$
Q_1(\nu,\alpha) = -\frac{1-\nu}{2\pi^2} i\sin\frac{\alpha}{2} \int_0^{\infty} \frac{\ln u}{(1+u)\cos\frac{\alpha}{2} + i(u+\nu)\sin\frac{\alpha}{2}} \cdot \frac{du}{1+u},
\tag{A.20}
$$

and then solve the integral using complex contour integration of the function $\ln^2 z / \left[ (1+z) \cos \frac{\alpha}{2} + i (z + v) \sin \frac{\alpha}{2} \right] (1+z)$. We eventually obtain

$$Q_1(v, \alpha) = \frac{\ln^2 \left[ e^{-i\alpha/2} \left( \cos \frac{\alpha}{2} + i v \sin \frac{\alpha}{2} \right) \right]}{4\pi^2}, \tag{A.21}$$

where the logarithm should be interpreted as belonging to the principal branch, $|\text{Im} \ln z| < \pi$. Expanding in powers of $\alpha$, we arrive at

$$Q_1(v, \alpha) = -\frac{(1-v)^2}{16\pi^2} \alpha^2 + \mathcal{O}(\alpha^3). \tag{A.22}$$

Finally, we estimate the contributions of terms of the form $\Upsilon_1(v, \alpha)$ that appear in the expression for the constant term in Eq. (32). Up to order $\alpha^2$ we have, according to Eqs. (33) and (A.18),

$$\Upsilon_1(v, \alpha) = \frac{1}{2\pi i} \int_v^1 \ln \frac{\Gamma \left( \frac{1}{2} + \frac{1}{2\pi i} \ln \left( \frac{1-x}{x-v} \right) \right)}{\Gamma \left( \frac{1}{2} - \frac{1}{2\pi i} \ln \left( \frac{1-x}{x-v} \right) \right)} \left[ \frac{i}{2} \alpha + \frac{x}{4} \alpha^2 \right] dx + \mathcal{O}(\alpha^3). \tag{A.23}$$

Employing a change of variables $\zeta = \ln \left( \frac{x-v}{1-x} \right)$ and the useful formula [103]

$$\ln \frac{\Gamma \left( \frac{1}{2} - iw \right)}{\Gamma \left( \frac{1}{2} + iw \right)} = -i \int_0^\infty \left[ 2w e^{-t} - \frac{\sin(wt)}{\sinh(t/2)} \right] \frac{dt}{t}, \tag{A.24}$$

one ends up with

$$\Upsilon_1(v, \alpha) = \frac{1-v}{8\pi} \int_{-\infty}^\infty d\zeta \frac{e^{-\zeta}}{(1+e^{-\zeta})^2} \left( 2i\alpha + \frac{1+ve^{-\zeta}}{1+e^{-\zeta}} \alpha^2 \right) \int_0^\infty \frac{dt}{t} \left[ \frac{\zeta}{\pi} e^{-t} - \frac{\sin \left( \frac{\zeta t}{2\pi} \right)}{\sinh(t/2)} \right] + \mathcal{O}(\alpha^3). \tag{A.25}$$

Switching the order of integration, the $\mathcal{O}(\alpha)$ term vanishes trivially due to the integrand being odd with respect to $\zeta$. Carrying out the integration of the rest, we arrive at

$$\Upsilon_1(v, \alpha) = -\frac{(1-v)^2}{16\pi^2} (1 + \gamma_E) \alpha^2 + \mathcal{O}(\alpha^3), \tag{A.26}$$

where $\gamma_E = \int_0^\infty \frac{e^{-t} + t - 1}{t(e^t - 1)} dt$ is the Euler-Mascheroni constant [108].

Adding up the different terms, we conclude that the expansion up to order $\alpha^2$ of the nonequilibrium deviation of the generating function (Eq. (37)) for $n = 1$ is given by

$$\begin{aligned}
\ln \frac{Z_1(\alpha)}{Z_1^{\text{eq}}(\alpha)} \approx & \frac{i}{2\pi} \left[ k_{F,R} - k_0 + \frac{1}{2} \int_{k_-}^{k_+} v(k) \, dk \right] \alpha L - \left( \int_{k_-}^{k_+} \frac{1 - v(k)^2}{16\pi} dk \right) \alpha^2 L \\
& + \left( 1 - \left( \frac{1 - v(k_-)}{2} \right)^2 - \left( \frac{1 + v(k_+)}{2} \right)^2 \right) \frac{\alpha^2 \ln L}{4\pi^2} \\
& + \frac{1 - v_0^2}{8\pi^2} \left( 1 + \gamma_E + \ln \left| 2 \sin \frac{\Delta k}{2} \right| \right) \alpha^2 \\
& - \left[ \frac{1 - v_0}{4\pi^2} \ln \left| \frac{\sin \frac{k_- + k_{F,R}}{2}}{\sin k_0} \right| + \frac{1 + v_0}{4\pi^2} \ln \left| \frac{\sin \frac{k_+ + k_{F,R}}{2}}{\sin k_0} \right| \right] \alpha^2 + \mathcal{O}(\alpha^3). \tag{A.27}
\end{aligned}$$

Applying Eq. (25), the expansion may be also expressed as

$$
\ln \frac{Z_1(\alpha)}{Z_1^{\text{eq}}(\alpha)} \approx -\frac{i}{2\pi} \left[ \int_{k_{F,R}}^{k_{F,L}} |r_R(k)|^2 \, dk \right] \alpha L - \left( \int_{k_-}^{k_+} \frac{|t_L(k) \, r_R(k)|^2}{4\pi} dk \right) \alpha^2 L
$$

$$
+ \frac{1}{4\pi^2} \left( 1 - \left| r_R\left(k_{F,R}\right) \right|^4 - \left| t_L\left(k_{F,L}\right) \right|^4 \right) \alpha^2 \ln L
$$

$$
+ \frac{|t_L(k_0) \, r_R(k_0)|^2}{2\pi^2} \left( 1 + \gamma_E + \ln \left| 2 \sin \frac{\Delta k}{2} \right| \right) \alpha^2
$$

$$
- \frac{|r_R(k_0)|^2}{2\pi^2} \ln \left| \frac{\sin k_{F,R}}{\sin k_0} \right| \alpha^2 + \mathcal{O}\left(\alpha^3\right). \tag{A.28}
$$

### A.4 The von-Neumman entanglement entropy

We present here details of the derivation of the asymptotic form for the unresolved vNEE in Eq. (44). From Eqs. (3) and (34), along with the fact that $Z_1 = 1$ by definition, we draw the following relation:

$$
\mathcal{C}_{\text{lin}} L + \mathcal{C}_{\text{log}} \ln L + \mathcal{C}_{\text{const}} = - \lim_{n \to 1} \left[ \partial_n \mathcal{I}_{\text{lin}}(n,0) L + \partial_n \mathcal{I}_{\text{log}}(n,0) \ln L + \partial_n \tilde{\mathcal{I}}_{\text{const}}(n,0) \right]. \tag{A.29}
$$

The explicit expression for $\mathcal{C}_{\text{lin}}$ in Eq. (45) is then straightforward to obtain. Calculating analytically the derivatives of $\partial_n \mathcal{I}_{\text{log}}(n,0)$ and $\partial_n \tilde{\mathcal{I}}_{\text{const}}(n,0)$, however, requires a more subtle analysis. The functions $Q_n(\nu,0)$ and $\Upsilon_n(\nu,0)$ that appear in those terms (defined in Eqs. (28) and (33)) have integral definitions with integrands that depend on $n$, and the integrals must be rewritten before one can estimate the required derivatives simply by taking the derivatives of the integrands.

Both $Q_n(\nu,0)$ and $\Upsilon_n(\nu,0)$ are defined as integrals over the interval $[\nu,1]$. By splitting it into the intervals $\left[\nu, \frac{1+\nu}{2}\right]$ and $\left[\frac{1+\nu}{2}, 1\right]$, and by changing variables to $u = \frac{x-\nu}{1-x}$ within the former and $u = \frac{1-x}{x-\nu}$ within the latter, we arrive at the following expressions:

$$
Q_n(\nu,0) = \int_0^1 \frac{dx}{2\pi^2 x} \left\{ \ln\left[ \left(1 + \frac{1+\nu}{2}x\right)^n + \left(\frac{1-\nu}{2}x\right)^n \right] + \ln\left[ \frac{\left(x + \frac{1+\nu}{2}\right)^n + \left(\frac{1-\nu}{2}\right)^n}{\left(\frac{1+\nu}{2}\right)^n + \left(\frac{1-\nu}{2}\right)^n} \right] \right\} - \frac{n}{12}, \tag{A.30}
$$

and

$$
\Upsilon_n(\nu,0) = \int_0^1 \frac{dx}{2\pi^2 x} \left\{ \ln\left[ \left(1 + \frac{1+\nu}{2}x\right)^n + \left(\frac{1-\nu}{2}x\right)^n \right] + \ln\left[ \frac{\left(x + \frac{1+\nu}{2}\right)^n + \left(\frac{1-\nu}{2}\right)^n}{\left(\frac{1+\nu}{2}\right)^n + \left(\frac{1-\nu}{2}\right)^n} \right] \right\}
$$

$$
\times \int_0^\infty \left[ \frac{\cos\left(\frac{\ln x}{2\pi}z\right)}{2\sinh\left(\frac{z}{2}\right)} - \frac{e^{-z}}{z} \right] dz - n\kappa_0, \tag{A.31}
$$

where the numerical constant $\kappa_0$ has been defined right after Eq. (43). In the derivation of Eq. (A.31), the formula of Eq. (A.24) was employed. The dependence on $n$ of the integrals featured in Eqs. (A.30) and (A.31) is manifested in terms of the same form, the corresponding

derivative of which is given by

$$
\lim_{n \to 1} \partial_n \left\{ \ln \left[ \left( 1 + \frac{1+\nu}{2} x \right)^n + \left( \frac{1-\nu}{2} x \right)^n \right] + \ln \left[ \frac{\left( x + \frac{1+\nu}{2} \right)^n + \left( \frac{1-\nu}{2} \right)^n}{\left( \frac{1+\nu}{2} \right)^n + \left( \frac{1-\nu}{2} \right)^n} \right] \right\}
$$

$$
= \frac{\left( 1 + \frac{1+\nu}{2} x \right) \ln \left( 1 + \frac{1+\nu}{2} x \right) + \frac{1-\nu}{2} x \ln x + \left( x + \frac{1+\nu}{2} \right) \ln \left( x + \frac{1+\nu}{2} \right)}{1 + x}
$$

$$
- \left( \frac{1+\nu}{2} \right) \ln \left( \frac{1+\nu}{2} \right). \tag{A.32}
$$

Finally, by recalling Eq. (25), we reach the results in Eqs. (46) and (47).

## A.5 Correlation matrix in a subsystem between two scatterers

We derive the approximate form appearing in Eqs. (58) and (59) for the correlation matrix of a subsystem situated between two scattering regions – region I on its left and region II on its right – assuming that the edges of the subsystem are far away from both regions. Energy eigenstates are given by scattering states, which for a wave incoming from the left take the form

$$
\langle n | \psi_k^{(L)} \rangle = \frac{1}{\sqrt{N}} \cdot \frac{t_L^{(I)}(k)}{1 - e^{2ik\ell} r_R^{(I)}(k) r_L^{(II)}(k)} \left[ e^{ikn} + e^{2ik\ell} r_L^{(II)}(k) e^{-ikn} \right], \tag{A.33}
$$

while for a wave incoming from the right,

$$
\langle n | \psi_k^{(R)} \rangle = \frac{1}{\sqrt{N}} \cdot \frac{t_R^{(II)}(k)}{1 - e^{2ik\ell} r_R^{(I)}(k) r_L^{(II)}(k)} \left[ r_R^{(I)}(k) e^{ikn} + e^{-ikn} \right], \tag{A.34}
$$

where $\ell$ is the number of sites between the scattering regions, and the convention $k > 0$ is used. By neglecting the effect of localized bound states as in Sec. 2, we may use Eqs. (A.33) and (A.34) in order to write the creation operator associated with site $n$ as

$$
a_n^\dagger = \sum_{k>0} \frac{1}{\sqrt{N}} \left( \frac{t_L^{(I)}(k)}{1 - e^{2ik\ell} r_R^{(I)}(k) r_L^{(II)}(k)} \right)^* \left[ e^{-ikn} + e^{-2ik\ell} r_L^{(II)}(k)^* e^{ikn} \right] a_{k,L}^\dagger
$$

$$
+ \sum_{k>0} \frac{1}{\sqrt{N}} \left( \frac{t_R^{(II)}(k)}{1 - e^{2ik\ell} r_R^{(I)}(k) r_L^{(II)}(k)} \right)^* \left[ r_R^{(I)}(k)^* e^{-ikn} + e^{ikn} \right] a_{k,R}^\dagger. \tag{A.35}
$$

By exchanging summation with integration and neglecting all Hankel terms, the elements of the correlation matrix can now be written as

$$
C_{mn} \approx \int_0^{k_{F,L}} \frac{dk}{2\pi} \left| \frac{t_L^{(I)}(k)}{1 - e^{2ik\ell} r_R^{(I)}(k) r_L^{(II)}(k)} \right|^2 \left[ e^{-i(m-n)k} + \left| r_L^{(II)}(k) \right|^2 e^{i(m-n)k} \right]
$$

$$
+ \int_0^{k_{F,R}} \frac{dk}{2\pi} \left| \frac{t_R^{(II)}(k)}{1 - e^{2ik\ell} r_R^{(I)}(k) r_L^{(II)}(k)} \right|^2 \left[ \left| r_R^{(I)}(k) \right|^2 e^{-i(m-n)k} + e^{i(m-n)k} \right]. \tag{A.36}
$$

Next, we employ the approximation

$$
\frac{1}{\left| 1 - e^{2ik\ell} r_R^{(I)}(k) r_L^{(II)}(k) \right|^2} \approx \frac{1}{1 - \left| r_R^{(I)}(k) r_L^{(II)}(k) \right|^2}, \tag{A.37}
$$

which is justified given that the difference between the two expressions oscillates as a function of $k$ with a frequency of $2\ell$, and thus after integration its contribution decays for $\ell$ that is large with respect to the Fermi wavelengths, $\left(k_{F,L}\right)^{-1}$ and $\left(k_{F,R}\right)^{-1}$. We therefore obtain

$$
\begin{aligned}
C_{mn} \approx & \int_{-k_{F,R}}^{-k_{F,L}} \frac{dk}{2\pi} \cdot \frac{\left|t_R^{(II)}(-k)\right|^2}{1 - \left|r_R^{(I)}(-k) \, r_L^{(II)}(-k)\right|^2} e^{-i(m-n)k} \\
& + \int_{-k_{F,L}}^{k_{F,R}} \frac{dk}{2\pi} e^{-i(m-n)k} + \int_{k_{F,R}}^{k_{F,L}} \frac{dk}{2\pi} \cdot \frac{\left|t_L^{(I)}(k)\right|^2}{1 - \left|r_R^{(I)}(k) \, r_L^{(II)}(k)\right|^2} e^{-i(m-n)k}.
\end{aligned} \tag{A.38}
$$

This result holds true regardless of the direction of the bias voltage. Once we explicitly distinguish between the two possible directions, we finally obtain the expression in Eqs. (58) and (59).

# B  Additional plots for the single impurity model

Focusing on the single impurity model defined in Sec. 4, the plots presented here illustrate the parameter dependence of the coefficients in the analytical asymptotics of the generating function (Eq. (34)). Fig. 9 demonstrates how, as $n$ increases, the dependence of $\mathrm{Re}\ln Z_n(\alpha)$ on $\alpha$ becomes flatter, which amounts to a narrower distribution of the corresponding charge-resolved Rényi moment about its peak. Figs. 10–11 depict the dependence of the coefficients on $\epsilon_0/t$ and $k_\pm$ (respectively), where the most conspicuous features are related to the divergences of the coefficients $\mathcal{I}_{\log}$ and $\tilde{\mathcal{I}}_{\mathrm{const}}$ that are discussed at length in Subsec. 3.2. In Figs. 10(b),(e)

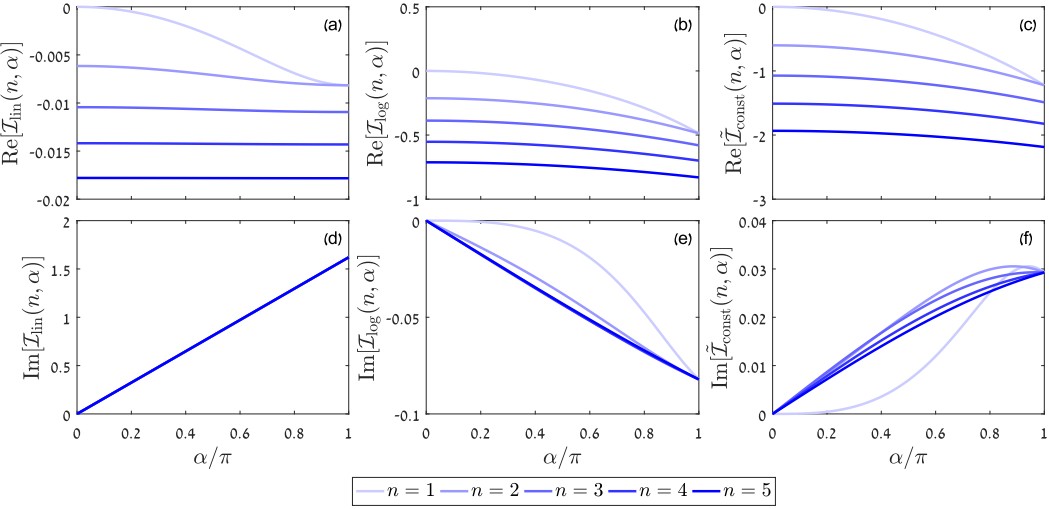

Figure 9: The single impurity model: Coefficients in the analytical asymptotic expression for $\ln Z_n(\alpha)$ in Eq. (34), as a function of $\alpha$ for different fixed values of $n$. The remaining parameters are set to $\epsilon_0 = t$, $k_{F,R} = \pi/2$ and $k_{F,L} - k_{F,R} = 0.1$. The top panels (a)–(c) show the real parts of the coefficients, and the bottom panels (d)–(f) show their imaginary parts. Panels (a) and (d) are for the coefficient $\mathcal{I}_{\mathrm{lin}}$, panels (b) and (e) are for $\mathcal{I}_{\log}$, and panels (c) and (f) are for $\tilde{\mathcal{I}}_{\mathrm{const}}$. Note that in panel (d) the different curves overlap because $\mathrm{Im}\left[\mathcal{I}_{\mathrm{lin}}\right]$ is dominated by the first term in Eq. (27), which is independent of $n$.

and 11(b),(e) the divergences of $\mathcal{I}_{\log}$ for $\alpha = \pi$ are related to points where either $v(k_-) = 0$ or $v(k_+) = 0$. In Figs. 10(c),(f) the coefficient $\tilde{\mathcal{I}}_{\text{const}}$ is seen to diverge for $\alpha = \pi$ and $v_0 = 0$. Note that in Fig. 11, the values of $\mathcal{I}_{\log}$ and $\tilde{\mathcal{I}}_{\text{const}}$ at the limit $\Delta k \to 0$ are fictitious since, as explained in Subsec. 3.2, throughout our calculations we implicitly assume $\Delta k \gg 1/L$.

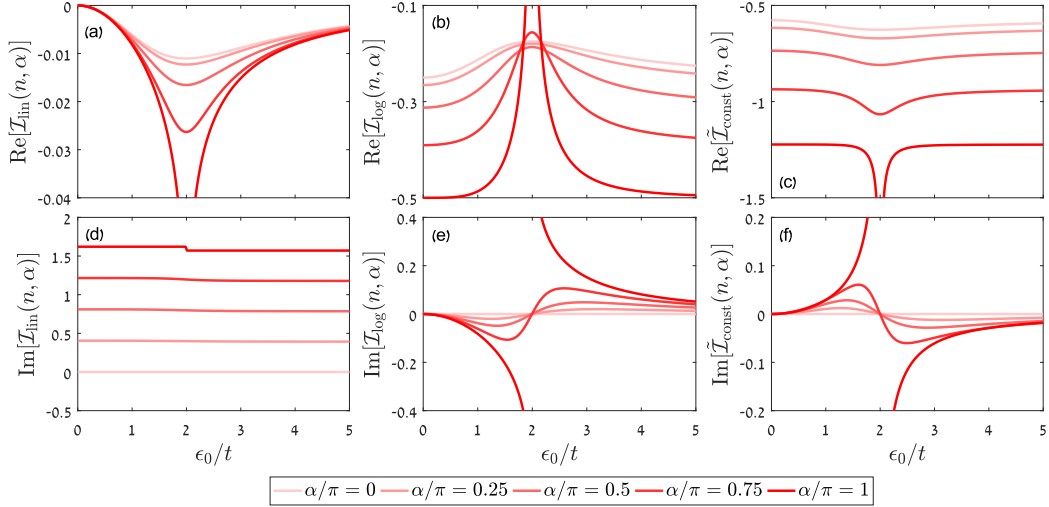

Figure 10: The single impurity model: Coefficients in the analytical asymptotic expression for $\ln Z_n(\alpha)$ in Eq. (34), as a function of $\epsilon_0/t$ for different fixed values of $\alpha$. The remaining parameters are set to $n = 2$, $k_{F,R} = \pi/2$ and $k_{F,L} - k_{F,R} = 0.1$. The top panels (a)–(c) show the real parts of the coefficients, and the bottom panels (d)–(f) show their imaginary parts. Panels (a) and (d) are for the coefficient $\mathcal{I}_{\text{lin}}$, panels (b) and (e) are for $\mathcal{I}_{\log}$, and panels (c) and (f) are for $\tilde{\mathcal{I}}_{\text{const}}$.

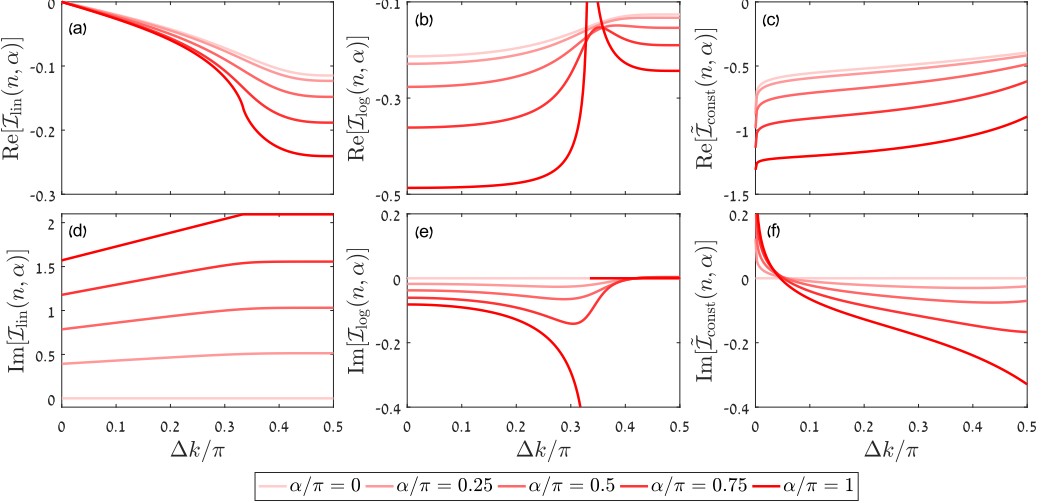

Figure 11: The single impurity model: Coefficients in the analytical asymptotic expression for $\ln Z_n(\alpha)$ in Eq. (34), as a function of $\Delta k = k_{F,L} - k_{F,R}$ for different fixed values of $\alpha$. The remaining parameters are set to $n = 2$, $\epsilon_0 = t$ and $k_{F,R} = \pi/2$. The top panels (a)–(c) show the real parts of the coefficients, and the bottom panels (d)–(f) show their imaginary parts. Panels (a) and (d) are for the coefficient $\mathcal{I}_{\text{lin}}$, panels (b) and (e) are for $\mathcal{I}_{\log}$, and panels (c) and (f) are for $\tilde{\mathcal{I}}_{\text{const}}$.

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
