# Peer review of "Entanglement Measures in a Nonequilibrium Steady State: Exact Results in One Dimension"

_SciPost Physics, doi:SciPost Phys. 11, 085 (2021)_

## Round 1 · Referee Report · Anonymous · 2021-5-17

Strengths
1) Interesting analytical results showing an unusual entanglement scaling.
2) First example of an entanglement equipartition-breaking term scaling with the subsystem size.
3) Comprehensive presentation of the results and the approximations.
Weaknesses
1) It is not present a discussion about the generality of the results.
Report
The manuscript "Entanglement Measures in a Nonequilibrium Steady State: Exact Results in One Dimension" by Shachar Fraenkel and Moshe Goldstein investigates entanglement entropies in a 1D lattice model, in particular a tight-binding chain containing a scattering region at its center to which a DC current voltage is applied.
The authors provide calculations of the generating function $Z_n(\alpha)$, to which Rényi Entropies, Von Neuman Entanglement Entropy and charge(symmetry)-resolved counterparts may be linked.
Under reasonable assumptions (large interval L in an infinite chain, distance from the scattering region d much larger than L) they obtain an approximate asymptotic expression for the generating function (Eq. 25) up to a logarithmic term in L. They appropriately highlight the relation between the terms in Eq.25 and the physical context, linking the volume term to the counting of the filled momentum states and the logarithmic term to the discontinuities of the energy distribution.
With the addition of the assumption of small bias they estimate an additional term which is independent of L (Eq.28).
The authors appropriately shed light on the incorrect behaviour of the analytical expression for $\alpha=\pm \pi$ and $\nu=0$. This implies that an analytical calculation of the charge(symmetry)-resolved Rényi moments cannot be carried out without a further approximation of the generating function, which however holds for large L. The calculation allow them to estimate an equipartition-breaking term scaling with the system size, first time observed in the literature.
Then the authors provide numerical simulations for the case of a scattering region consisting of a single site and test the validity of the approximations computing $Z_n(\alpha)$ in several conditions.
Eventually they generalize their calculation to the case of multiple scatterers.
The paper is well written and it clearly shows the effort of the authors in detailing all the criticalities of the calculations. They provide a comprehensive derivation of the analytical results in the appendices, if not present in the main text, and investigate the validity of the approximations extensively in section 4.3.
The result is innovative and well presented. They show an unusual scaling of entanglement with both a volume-law and logarithmic term and provide the first example, at the best of my knowledge, of an equipartition-breaking term scaling with the system size.
The findings presented here could be a useful addition to the topic of entanglement scaling out-of-equilibrium as well as to the newly popular topic of charge(symmetry)-resolved entanglement . In conclusion, I recommend the paper for publication.
Nevertheless, a short list of more specific comments/questions follows.
Requested changes
1) Right after Eq.15, the authors write the same piecewise function $\tau(k)$ in the case $k_{F,R}>k_{F,L}$. It might have been easier for the reader if it was written, as Eq.15, in the equation environment.
2) Similarly Eq.23 might be written in the same fashion of Eq.15 and the explanation before simplified.
3) The authors claim that the scaling law they write in Eq.42 should be more common in nonequilibrium setups, nonetheless confirming that they deal with a specific class of steady-states, for which analytical calculations are possibile ("excited eigenstates of an Hamiltonian with Fermi-discontinuous distribution of the excitations"). It would be interesting a comment on this statement, as it is not evident which are the non-equilibrium setups they refer to. Are open systems counted among these, e.g. scattering regions given by local pump/loss terms?
4) The authors find an anti-symmetric equipartition-breaking term and highlight the fact that it is odd with respect to $Q_A-\langle Q_A \rangle$. It seems it is related to the fine-tuned setting. Do they have any particular insight on how general this odd symmetry breaking is and if it could be seen in other contexts?
Author: Shachar Fraenkel on 2021-08-29 [id 1718]
(in reply to Report 1 on 2021-05-17)
We would like to thank the Referee for reading our manuscript thoroughly, and for the comments which gave us the opportunity to clarify some important points within the text. We are very glad that the Referee has recommended our manuscript for publication in SciPost Physics. We detail below our replies to specific comments by the Referee; our answers to the last two comments should clarify the generality of our results, and alleviate the weakness noted by the Referee in this regard.
1) The Referee writes: “Right after Eq.15, the authors write the same piecewise function $\tau\left(k\right)$ in the case $k_{F,R}>k_{F,L}$. It might have been easier for the reader if it was written, as Eq.15, in the equation environment.”
We added Eq. (16) per the suggestion of the referee.
2) The Referee writes: “Similarly Eq.23 might be written in the same fashion of Eq.15 and the explanation before simplified.”
We added Eq. (24) to simplify the explanation.
3) The Referee writes: “The authors claim that the scaling law they write in Eq.42 should be more common in nonequilibrium setups, nonetheless confirming that they deal with a specific class of steady-states, for which analytical calculations are possibile ("excited eigenstates of an Hamiltonian with Fermi-discontinuous distribution of the excitations"). It would be interesting a comment on this statement, as it is not evident which are the non-equilibrium setups they refer to. Are open systems counted among these, e.g. scattering regions given by local pump/loss terms?”
The attributes of the steady state which are responsible for the scaling law are: (i) The existence of a set (of non-vanishing measure) of single-particle states with occupation probabilities which are neither 0 nor 1; this gives rise to the extensive contribution, proportional to the subsystem size. (ii) The presence of Fermi discontinuities, which are the origin of the term logarithmic in the subsystem size. These attributes potentially apply to a large variety of nonequilibrium models at zero temperature, including, e.g., a chain with scattering regions involving particle pumping or loss, as suggested by the Referee. It may also apply to Dirac fermions in the presence of impurities and bias voltage, as suggested by the other Referee. We expanded the second paragraph of Section 6 accordingly.
4) The Referee writes: “The authors find an anti-symmetric equipartition-breaking term and highlight the fact that it is odd with respect to $Q_{A}-\langle Q_{A}\rangle$. It seems it is related to the fine-tuned setting. Do they have any particular insight on how general this odd symmetry breaking is and if it could be seen in other contexts?”
Since the model we have studied is, to the best of our knowledge, the first to feature an asymmetric equipartition-breaking term, it is difficult to assess the generality of this particular result. Nevertheless, we know that it did not emerge in previously studied nonequilibrium models, cf. Ref. [73] (according to the numbering in the resubmitted version) where symmetry-resolved entanglement is studied following a symmetric quench. The steady state in Ref. [73], however, does not carry a net current of particles, as opposed to the steady state that we have studied. As such a net current (by definition) breaks inversion symmetry in the system, one may speculate that the odd equipartition-breaking term is related to it. We added a sentence to Section 6 which emphasizes this point.
Author: Shachar Fraenkel on 2021-08-29 [id 1717]
(in reply to Report 2 on 2021-08-06)We would like to thank the Referee for reading our manuscript thoroughly, and for the comments which gave us the opportunity to clarify some important points within the text. We are very glad that the Referee has recommended our manuscript for publication in SciPost Physics. We detail below our replies to specific comments by the Referee.
1) The Referee addressed several typos which were all fixed.
2) The Referee writes: “In section 4.3, the authors discuss the effect of a finite distance $d$ between the subsystem and the impurity and they conclude that their results are reliable until $d \gg L$. Could they comment about what would change if this hypothesis does not hold and how this would affect the main results of this work? Could they also comment about what would change considering an impurity inside the subsystem?" Since the distance $d$ of the subsystem from the impurity amounts to a boundary condition, it is not expected to affect extensive properties of the subsystem. In particular, the leading term of the generating function (and therefore also of the Renyi moments and the vNEE), which is linear in $L$, should not depend on $d$, regardless of the size of $d$ relative to $L$. The term which is logarithmic in $L$, on the other hand, is generally sensitive to boundary conditions, and therefore may depend on $d$ for any $L$. Similar considerations should apply to a subsystem with the impurity in its interior. We expanded Section 4.3 to address this issue, and also changed one sentence in Section 6 accordingly.
3) The Referee writes: “Do they have any insight if a similar treatment for the massless Dirac field on the line in the presence of a point-like defect could be interesting? Namely, if a field theory analysis of a similar problem could show the main findings presented here. From this observation, one could wonder whether the equipartition-breaking terms could be also a fingerprint of the fact that the presence of a defect breaks the translational symmetry of the model and, as a consequence, the underlying conformal invariance which is usually associated with the equipartition of entanglement in presence of an abelian symmetry.” The main findings of our work all stem from the form of the two leading (linear and logarithmic in $L$) terms in the expression for the generating function; these terms result, respectively, from having single-particle states with occupation which is neither 0 nor 1, and from having Fermi discontinuities. Both features are consequences of having a finite voltage bias. We therefore expect the main findings to apply also to a 1D massless Dirac field in the presence of an impurity and a voltage bias, as the Referee suggested. Indeed, for our model in the limit of a small bias voltage one may linearize the dispersion relation, so that the single-particle states within the voltage window are effectively described as those of a free massless Dirac fermion, confirming this expectation in that limit. We expanded the second paragraph in Section 6 to address this point.

---

## Round 1 · Referee Report · Anonymous · 2021-8-6

Strengths
1) Nontrivial results on entanglement measures and their symmetry resolution.
2) The paper is written in a clear way and the analysis of the results found is extremely detailed
Report
This paper presents exact results for the entanglement measures of a contiguous subsystem of a tight-binding chain with general scattering regions subject to a voltage bias at zero temperature. This leads to a current-carrying steady state which is still a pure state. Within this setup, the authors find new results concerning the generating function, $Z_n(\alpha)$, defined in Eq. (17), from which they can compute both the entanglement entropies and their decomposition in the charge sectors of the model. The main tool used to perform the aforementioned analysis is the generalized Fisher-Hartwig conjecture: it allows them to compute systematic expansions of these entropies which provide exact expressions up to constant terms in the subsystem size, $L$. The analytical expression found turns out to be singular at $\nu=0,\alpha=\pm \pi$, while whenever $\nu$ is finite the divergence at $\alpha =\pm \pi$ eliminates the ones already found in literature for $\log Z_n(\alpha)$. The analytical results are also confirmed by the numerical results which are widely discussed in Section 4.
The novelties found can be summed up as follows:
- a combination of volume-law and logarithmic contributions to the entanglement, whose physical meaning can be traced back to the momentum eigenstates within the bias voltage window and to the discontinuities of the Fermi-Dirac distribution, respectively;
- deviations from equipartition of the symmetry-resolved entropies which scale as the square root of the subsystem size.
The paper is well-written and it contains some nontrivial results on a recently popular topic. The calculations are clearly explained and detailed, especially in the appendices. Moreover, the authors compare their results with the existing literature throughout, also stressing how they contribute to generalize some of them. Therefore, I recommend the paper for publication.
Here is nevertheless a short list of comments/questions/typos:
- Page 2: d being the spacial dimension $\rightarrow$ d being the spatial dimension;
- Caption Fig.5: "broken lines" could be replaced by "dashed lines";
- Page 23: has reduced to problem $\rightarrow$ has reduced the problem;
- Page 25: leading equipartiton-breaking $\rightarrow$ leading equipartition-breaking;
- In section 4.3, the authors discuss the effect of a finite distance $d$ between the subsystem and the impurity and they conclude that their results are reliable until $d \gg L$. Could they comment about what would change if this hypothesis does not hold and how this would affect the main results of this work? Could they also comment about what would change considering an impurity inside the subsystem?
- Do they have any insight if a similar treatment for the massless Dirac field on the line in the presence of a point-like defect could be interesting? Namely, if a field theory analysis of a similar problem could show the main findings presented here. From this observation, one could wonder whether the equipartition-breaking terms could be also a fingerprint of the fact that the presence of a defect breaks the translational symmetry of the model and, as a consequence, the underlying conformal invariance which is usually associated with the equipartition of entanglement in presence of an abelian symmetry.

---

## Round 2 · Referee Report · Anonymous (Referee 2) · 2021-8-30

Report

I am happy with the changes/clarifications made by the authors and I recommend this article for publication.

---

## Round 2 · Referee Report · Anonymous (Referee 1) · 2021-9-12

Report

The authors answered all the points raised in the first round of review.

I thank them for their clear reply and raccomend the paper for publication.

---

## Round 2 · Author Response

Dear Editor,

Attached is a new version of our manuscript following the requested revision. We would like to thank the Referees for reading our manuscript thoroughly, and for their comments which gave us the opportunity to clarify some important points within the text. We are very glad that both Referees have recommended our manuscript for publication in SciPost Physics. We detail in the response to each referee the modifications we have applied following each specific comment. In addition, we have used the opportunity to correct some typos and add a few additional references. The parts of the manuscript that were modified in light of these comments by the Referees are marked in red in the new manuscript. We believe that this revised version is now ready for publication.

Sincerely,
Shachar Fraenkel and Moshe Goldstein

---

## Round 2 · List of Changes

The changes to our manuscript are detailed in our responses to the Referees.

---

## Editorial Decision

published